# Preliminary Study on Light-Activated Antimicrobial Agents as Photocatalytic Method for Protection of Surfaces with Increased Risk of Infections

**DOI:** 10.3390/ma14185307

**Published:** 2021-09-14

**Authors:** Razvan Bucuresteanu, Lia-Mara Ditu, Monica Ionita, Ioan Calinescu, Valentin Raditoiu, Bogdan Cojocaru, Ludmila Otilia Cinteza, Carmen Curutiu, Alina Maria Holban, Marius Enachescu, Laura-Bianca Enache, Gabriel Mustatea, Viorel Chihaia, Adela Nicolaev, Elena-Larisa Borcan, Grigore Mihaescu

**Affiliations:** 1Department of Microbiology, Faculty of Biology, University of Bucharest, Intr. Portocalelor no 1-3, 060101 Bucharest, Romania; razvan.bucuresteanu@drd.unibuc.ro (R.B.); carmen.curutiu@bio.unibuc.ro (C.C.); alina.m.holban@bio.unibuc.ro (A.M.H.); grigore.mihaescu@bio.unibuc.ro (G.M.); 2Faculty of Biology, Research Institute, University of Bucharest, Soseaua Paduri 90-92, 50663 Bucharest, Romania; 3Faculty of Applied Chemistry and Materials Science, University Politehnica of Bucharest, Splaiul Independenței no 313, 060042 Bucharest, Romania; ionita_monica@yahoo.com (M.I.); ioan.calinescu@upb.ro (I.C.); 4Laboratory of Functional Dyes and Related Materials, National Institute for Research & Development in Chemistry and Petrochemistry—ICECHIM, 202 Splaiul Independentei, 6th District, 060021 Bucharest, Romania; vraditoiu@icechim.ro; 5Department of Organic Chemistry, Biochemistry & Catalysis, Faculty of Chemistry, University of Bucharest, Bdul Regina Elisabeta 4-12, 030016 Bucharest, Romania; bogdan.cojocaru@chimie.unibuc.ro; 6Department of Physical Chemistry, Faculty of Chemistry, University of Bucharest, Bdul Regina Elisabeta 4-12, 030016 Bucharest, Romania; ocinteza@gw-chimie.math.unibuc.ro; 7Center for Surface Science and Nanotechnology, University Politehnica of Bucharest, 313 Splaiul Independentei, 060042 Bucharest, Romania; marius.enachescu@cssnt-upb.ro (M.E.); laura.bianca@cssnt-upb.ro (L.-B.E.); 8Academy of Romanian Scientists, 54 Spaiul Independentei, 050094 Bucharest, Romania; 9National R&D Institute for Food Bioresources—IBA Bucharest, 5 Ancuţa Băneasa Street, 020323 Bucharest, Romania; gabi.mustatea@bioresurse.ro; 10Institute of Physical Chemistry “Ilie Murgulescu”, Romanian Academy, Splaiul Independentei 202, 060021 Bucharest, Romania; vchihaia@icf.ro; 11Department of Surfaces and Interfaces, National Institute of Materials Physics, Atomistilor 405A, 077125 Magurele, Romania; adela.nicolaev@infim.ro (A.N.); elena-larisa.borcan@drd.unibuc.ro (E.-L.B.); 12Faculty of Physics, University of Bucharest, Atomistilor 405, 077125 Magurele, Romania

**Keywords:** light-activated antimicrobial agents, healthcare-associated infections, antimicrobial strategies, copper-doped TiO_2_ anatase pigment

## Abstract

Preventing and controlling the spread of multidrug-resistant (MDR) bacteria implicated in healthcare-associated infections is the greatest challenge of the health systems. In recent decades, research has shown the need for passive antibacterial protection of surfaces in order to reduce the microbial load and microbial biofilm development, frequently associated with transmission of infections. The aim of the present study is to analyze the efficiency of photocatalytic antimicrobial protection methods of surfaces using the new photocatalytic paint activated by light in the visible spectrum. The new composition is characterized by a wide range of analytical methods, such as UV-VIS spectroscopy, electron microscopy (SEM), X-ray powder diffraction (PXRD) or X-ray photoelectron spectroscopy (XPS). The photocatalytic activity in the UV-A was compared with the one in the visible light spectrum using an internal method developed on the basis of DIN 52980: 2008-10 standard and ISO 10678—2010 standard. Migration of metal ions in the composition was tested based on SR EN1186-3: 2003 standard. The new photocatalytic antimicrobial method uses a type of photocatalytic paint that is active in the visible spectral range and generates reactive oxygen species with inhibitory effect against all tested microbial strains.

## 1. Introduction

The COVID-19 pandemic has demonstrated the vulnerabilities of modern society and showed how little is known about medical ethics and public infection control strategies [1,2]. One of the greatest challenges of the health systems, therefore, remains the problem of healthcare-associated/acquired infections (HAI) that cause, every year, 4.5 million infections in Europe, leading to significant mortality and financial losses for health systems [3]. The first two main problems that directly influence the incidence of HAIs are multidrug-resistant (MDR) bacteria spread and persistence [4,5] and the long-time colonization of the medical surfaces with highly pathogenic strains that generate mono or multispecies biofilms [6,7,8]. Pathogens such as vancomycin-resistant enterococci (VRE), methicillin-resistant *Staphylococcus aureus* (MRSA), multiresistant Gram-negative bacilli (*Acinetobacter boumanii*, *Pseudomonas aeruginosa*, *Klebsiella* sp.), norovirus, and *Clostridium difficile* persist in the health care environment for days and increase the risk of acquisition for other patients [9]. The most important specialists in the field have drawn attention to the fact that we are heading towards a global “post-antibiotic” era and a future pandemic crisis [10].

The patient care areas, all touchable surfaces in the hospital but also indoor air, are the most important reservoirs of pathogens, with direct implications in the transmission pathway of infections from one person to another [11]. Active methods of disinfection and cleaning that follow a written specification of cleaning services are often inefficient in the mechanical and chemical removal of pathogens, especially the MDR strains. An Australian study demonstrated persistent MRSA contamination after using hydrogen peroxide [12,13]. Moreover, there are reports showing that surfaces remain contaminated with VRE when there is an inadequate contact time between a surface and applied disinfectant, and when items or surface are sprayed and wiped over, rather than being actively scrubbed [14]. Those are a few reasons that lead to a practice idea of passive antimicrobial protection measures, with permanent action that can reduce the possibility of biofilm formation and MDR bacteria spread, followed by the decrease of the microbial load on the contact surfaces [15,16,17]. 

Passive protection measures are implemented in the form of antimicrobial and antiadhesive coatings on various critical surfaces [10]. Very good results have been obtained by applying these antimicrobial and antiadhesive coatings on the medical supplies surfaces or on constructed surfaces that can be contaminated with pathogens [18]. There are four main strategies for obtaining antimicrobial surfaces: (i) the release of antimicrobial agents (depending on leaching of incorporated antimicrobial agents); (ii) mechanical damage of the membrane of the pathogens by contact (the cell membranes are disrupted by the contact with immobilized compounds); (iii) use of low-energy surfaces to inhibit the microbial adherence and biofilm development to the surface (immobilization of molecules such as polyethylene glycol and zwitterion that can inhibit cell wall protein adsorption); (iv) use of light-activated coatings (generation of the reactive oxygen species (ROS) such as singlet oxygen and hydroxyl radicals by photosensitizers that can degrade microbial pathogens, causing damage to their DNA or to cell membrane) [19]. 

The persistent antimicrobial properties have potential in reduction of microbial load on clinical settings surfaces [20,21]. However, these coatings have not become widespread in the healthcare environment due to unknown emissions and ecotoxicological effects [22]. There are no risk-benefit analyses, nor a unitary regulatory system approved by regulators, and the lack of clear evidence of benefits and risks can lead to the loss of the potential of these hedging models [20,23,24,25,26].

New strategies for preventing and fighting healthcare-associated infections consider the persistence of pathogens in biofilms and prevent their adherence to different surfaces [27]. At present, a series of biocidal compositions are used with real success, in the form of washable paints, protective resins, and fabrics and linen, in which antimicrobial agents of the type 1,2-Benzothiazol-3(2H)-one (CAS no. 2634-33-5), triclosan (CAS no. 3380-34-5), various polymers, copper, or silver are incorporated [28]. Although these measures have proved their usefulness, they have several shortcomings caused by the active biotoxic agents contained in the matrix of these compositions.

The main problem is the sublimation of active biotoxic agents in the composition matrix, because they are either active eluting agents (e.g., ions or nanoparticles of silver, copper, zinc, or antibiotics, chloride, iodine, etc.) or immobilized molecules that become active on contact (e.g., quaternary ammonium polymers or peptides) [29]. As a result, the biocidal effect of these compositions decreases with application and soon becomes ineffective in controlling pathogens.

The second major problem is related to the high toxicity of active biotoxic agents. Sublimation produces toxic fumes that can be inhaled. Numerous cases of skin allergies caused by prolonged exposure to 1,2-Benzothiazol-3(2H)-one (CAS no. 2634-33-5) have been reported [28,30]. There is a call for the European control authorities to limit the use of 1,2-Benzothiazol-3(2H)-one (CAS no. 2634-33-5) in various products, in order to eliminate the risk of skin allergies in antimicrobial compositions. Silver has been used since antiquity as an antibacterial product. The new types of silver-polymer composites are based on the incorporation of silver nanoparticles in different substrates. It is known that there is a microbial resistance to silver, especially Gram-negative bacteria, and not Gram-positive bacteria. This resistance to silver may be genetically encoded in chromosomes or in transferable plasmids to other bacteria [31].

Achieving antibacterial protection of surfaces exposed to the risk of contamination with pathogens required the production of photocatalytic antimicrobial paint that can be used as an ecological coating solution with high long-term efficiency. This is a versatile solution that ensures the decoration of various surfaces. At the same time, it offers a practical antimicrobial solution that does not allow the adherence of pathogens to different surfaces.

In recent years, it has been shown that photocatalysis plays an important role in environmental decontamination processes [32] based on the chemical reactions triggered by the interaction of a photosensitive substance, called photosensitizer [33]. Titanium dioxide (TiO_2_) can be used as a catalyst that absorbs light of different wavelengths, owing to its electronic structure, and it is attractive for photocatalytic microbicidal activity because of its relatively low cost, natural abundance, and superior chemical stability [32].

TiO_2_, generally known as titania, is an n-type semiconductor due to the presence of oxygen vacancies [34]. It has been shown that the photocatalytic activity of TiO_2_ is influenced by the crystal structure, the specific surface, the particle size distribution, the porosity, or the density of the surface hydroxyl groups [35]. These characteristics influence the electron-hole pairs production, the adsorption-desorption on the surface, and the redox processes.

When irradiating TiO_2_ with UV radiations, pairs of holes in the valence band and electrons in the conduction band are created. For TiO_2_, the valence band is represented by the 2p orbitals of oxygen—highest occupied molecular orbital (HOMO) orbitals—and the valence band consists of free 3D orbitals of the titanium atom—lowest unoccupied molecular orbital (LUMO) orbitals. There is an energy difference of 3.2 eV between the two bands, a difference that forms the band-gap (the forbidden band)—energy corresponding to electromagnetic radiation with a wavelength of 360–370 nm.

It is known that if transitional metals are introduced in the crystalline structure of TiO_2_, the energy levels of TiO_2_ change and the consequence is the decrease of the band-gap [33]. The replacement of titanium with transition metal ions leads to the formation of a new energy level between the HOMO and LUMO band of the TiO_2_ crystal. The new energy level introduced by the dopant in the TiO_2_ network will capture the excited electrons from the HOMO band of TiO_2_, will suppress the recombination of charge carriers, and will be a trap for the electron-hole pairs, thus allowing the transport of several carriers to the surface. As a consequence, the energy level of the dopant will decrease the energy level of the TiO_2_ band-gap and will allow the photoexcitation of TiO_2_ with electromagnetic radiation from the visible spectrum.

The holes from the valence band react with water molecules or hydroxyl ions to form hydroxyl radicals which are very strong oxidants of organic molecules. It has been confirmed that oxygen vacancies can be treated as electron donors that determine the “n” type conductivity. Oxygen adsorption has a large influence on photoconductivity in porous nanocrystalline TiO_2_. The generated electron-hole pairs react with water or oxygen to produce radicals (Ox) on the surface of the semiconductor. 

Photocatalytic activity can be enhanced by suppression of electron-hole recombination. The most important method of suppression is to load the surface of TiO_2_ particles with metals.

As organic species adsorb and desorb from the surface of the catalyst, the radicals formed react easily with the adsorbed species, oxidizing them, forming CO_2_, H_2_O, and partially oxidized species [35].

Studying the mechanism of this reaction, Philippopoulos and Nikolaki [34] presented the most important stages of the process as the following:

Generation of charge carriers (electron-hole pairs):TiO_2_ + hν → h^+^ + e^−^

Capture of charge carriers:h^+^ + Ti(IV)OH → Ti(IV)OH^+^
e^−^ + Ti(IV)OH → Ti(III)OH
e^−^ + Ti(IV) → Ti(III)

Charge recombination:e^−^ + Ti(IV)OH^+^ → Ti(IV)OH
h^+^ + Ti(III)OH → Ti(IV)OH

Charge transfer to the interface region (where P is the organic pollutant and P* represents the oxidized form of P) [35]:e^−^ + Ti(IV)OH^+^ + P → Ti(IV)OH + P*
Ti(III)OH + O_2_ → Ti(IV)OH + O_2_^−^

The disadvantage of UV-A photocatalytic activation of TiO_2_, but also the benefit, is that, however, the photocatalytic reactions of TiO_2_, when induced, have led to the development of laboratory methods to obtain doped TiO_2_ that can be excited by electromagnetic radiation, generally in the 450–500 nm spectral range (non-toxic visible range). However, the problem of the doped TiO_2_ particle size remains. All currently known laboratory methods succeed in obtaining doped TiO_2_ nanoparticles [29,32]. However, industrially, the nanoparticles are used only in the manufacture of electronic components, under special technical conditions. The dyes and paints industry, as well as the consumer goods industry, are not in agreement with nanoparticles for two reasons: (i) being extremely light, they could persist in the air for a long time and do not deposit, causing serious occupational diseases. The European Union has regulated the industrial use of nanoparticles and imposed the use of extremely expensive personnel protection systems, but it has also required that industrial pigments such as TiO_2_ must be certified to be larger than nanoparticles; (ii) nanoparticles introduced in different coating formulations have the ability to sublimate and because of this, the concentration of nanoparticles in different formulations decreases over time and soon reaches the limit of minimum inhibitory concentrations, which leads to loss of their functional role.

The aim of the present study was to study and confirm a new method of photocatalytic protection of surfaces by using light-activated photocatalytic antimicrobial paint active in the visible spectrum that contains copper-doped TiO_2_ anatase, in order to evaluate and certify photocatalytic products for industrial production. It is known that the TiO_2_ excitation with respect to the photocatalytic reaction is initiated only by UV-A light (370–380 nm) [34]. Compared to the conventional photocatalytic processes, the excitation of the doped TiO_2_ pigment from the proposed product occurs on the entire visible spectrum (400–700 nm), hypothesis confirmed by the experimental data presented in the present paper. The antimicrobial effect is based on the generation of local, exogenous reactive oxygen species (ROS), with action on multiple cellular targets, generating damage in the microbial cell wall (Figure 1). 

Two major benefits were obtained by this photocatalytic antibacterial method: (i) the possibility of maintaining the antimicrobial and antibiofilm effect during the presence of patients in health facilities; (ii) the method does not allow the selection of resistant bacterial cells and the inhibitory effect can be produced for a longer period. Microorganisms do not possess specific defense mechanisms against exogenous ROS species, with no possibility of developing resistance to these biotoxic agents.

## 2. Materials and Methods

### 2.1. Preparation of Copper-Doped TiO_2_ Pigment and Photocatalytic Antimicrobial Washable Paint ALINNA

Copper-doped TiO_2_ pigment and photocatalytic antimicrobial paint were prepared based on the methodology described in Romanian patent application A/00297/2020 [36], as well as in patent RO 132438 B1 (PCT WO 2019/074386 A1) [37]. The materials used were for industrial use, certified as such. The manufacturer where the small production batch was manufactured has implemented and certified a quality and environmental management system in accordance with ISO 9001: 2015 [38] and ISO 14001: 2015 [39] (ICT certificate 15 100 1910630 and 15 104 191712 issued by TÜV Thüringen e.V.). The batch obtained was approved internally in the quality control laboratory of the respective unit, laboratory certified by TÜV Thüringen e.V.

### 2.2. Evaluation of the Physical Properties

For the new photocatalytic antimicrobial paint, the following physical parameters were measured: the paint density determined by the pycnometer method using Ascott Analytical pycnometer, with the reference color 1.60–0.05 in accordance with the working instruction elaborated by SR EN ISO—2811-1-2002 [40]; Brookfield viscosity determined using Fungilab™ Smart Series L Model Rotational Viscometer, with rotor 6, 20 rpm at 20 °C, in accordance with working instructions developed by ISO 2884-2: 2003 [41]; drying time determined based on ISO 1517/1999 [42]; the washability of the new paint tested by determining the wet abrasion resistance (200 cycles), test methodology developed based on EN ISO 11998: 2007 [43].

### 2.3. Evaluation of UV-Vis Optical Absorption Spectra

The spectra were collected under ambient conditions using Specord 250 equipment (Analytik Jena AG, Jena, Germany), in the 300–1100 nm range (Δλ = 2 nm, scanning speed = 10 nm/s, integration time = 0.2 s). The equipment included an integrating sphere as measuring device in reflectance mode. MgO was used as reference material.

### 2.4. X-ray Powder Diffraction (PXRD)

X-ray powder diffraction (PXRD) was collected at room temperature using a Rigaku SmartLab, Tokyo, Japan system that uses monochrome radiation Cu kα (λ =1.5406 Å, 40 kV, 200 mA), with a scanning rate of 2 degrees per minute, in the range 2θ = 5–100 degrees.

### 2.5. Scanning Electron Microscopy (SEM) and Energy-Dispersive X-ray (EDX)

The surface morphology and elemental composition of the powder were studied by scanning electron microscopy (SEM) (Hitachi SU 8230, Hitachi High-Tech Corporation, Tokyo, Japan) equipped with an energy-dispersive X-ray detector (EDX, Oxford Instruments, Oxford, UK).

### 2.6. X-ray Photoelectron Spectroscopy (XPS)

The measurements were performed in an ultrahigh vacuum chamber (base pressure ~1 × 10^−8^ Pa) manufactured by Kratos Analytical, UK, at room temperature. This is an automated installation for XPS coupled to a reaction cell for studies of surface reactions at high temperatures and pressures (1000 °C, 4 × 10^5^ Pa), with gas cabinet with four ways. The excitation sources are monochromatic Al Kα (1486.7 eV) and dual Al Kα (1486.7 eV)/Mg Kα (1253.6 eV) anode.

### 2.7. Specific Migration Tests of Heavy Metals

Specific migration tests were performed by inductively coupled plasma mass spectrometry (ICP-MS) using a NexION300Q equipment with triple cone interface and quadrupole ion deflector. Parameters of the equipment used in analysis are presented in Table 1.

### 2.8. Photocatalytic Absorption Tests for 2% Copper-Doped TiO_2_ Pigments 

The tests were performed using a JASCO V570 spectrophotometer. An opaque wall test chamber of its own design was used (Figure 2).

For testing in the ultraviolet field, a UV lamp OSRAM HQE 40 was used (emission spectrum in the range 300 nm ≤ λ ≤ 420 nm, with an irradiance E = (20 ± 0.5) W/m^2^ (measured on the tested sample). For testing in the visible range, LED projectors (emission spectrum exclusively in the range of 400 nm ≤ λ ≤ 800 nm, with an irradiance E = (15 ± 0.5) W/m^2^ (measured at the level of the tested sample) were used. The emission spectrum is shown in Figure 3.

For testing in the UV-Vis domain (simulated sunlight) an ATLAS NXe 2000 HE xenon arc lamp was used (emission spectrum exclusively in the range 300 nm ≤ λ ≤ 800 nm, with an irradiance E = (42 ± 0.5 W/m^2^) (measured at the level of the test sample)). For the test, a quantity of m = 0.07 ± 0.01 g photocatalytic antimicrobial paint ALINNA was deposited on the microscope slides, with a layer thickness g = 0.03 ± 0.005 mm, and the area covered was A = 16 ± 0.5 cm^2^. The photochemical conversion obtained in mole units per square meter and hour (mole/(m^2^h)) was measured. All devices that encounter methylene blue solutions (MBs) are made of materials that negligibly adsorb MB (glass, stainless steel, polyethylene, polypropylene, polyacrylate, or certain low-emission silicones). The experiment was performed considering that the scattered light should be as low as possible. Aqueous MB solutions with dye concentration in the Beer–Lambert field were used for testing. The initial concentration of MB for the test solution was c_0_ = (20 ± 0.1) µmol/L and the conditioning solution must have the same concentration c = (20 ± 0.1) µmol/L. The exposure was performed using 20 mg/L MB water solution with volume V = 45–75 mL, a photocatalytic coating surface A = 16 cm^2^, a source of irradiation depending on the spectral range for which the evaluation of the photocatalytic activity and a length of the measuring layer d = 10 mm were made. 

The absorbance of the methylene blue solution was measured at a wavelength λ = (664 ± 2) nm. The absorbance was determined by external measurements of the test solution and then the return of the measured solutions to the test solution. The absorbance was measured at 30 min intervals, at most until the solution discolored. The molar absorption coefficient of the MB solution considered was ε = 7402.8 m^2^/mol. A UV/Vis spectrometer (Jasco UV-Vis-NIR V-570) was used to determine the MB concentration, calibrated in the measuring range 400 nm ≤ λ ≤ 800 nm.

### 2.9. Antimicrobial Tests

The antimicrobial assays were performed using standard microbial strains that were included in the microbial collection of University of Bucharest, Faculty of Biology, Microbiology Department: *Staphylococcus aureus* ATCC 25923, *Pseudomonas aeruginosa* ATCC 27853, and *Candida albicans* ATCC 10231. For experiment assay, two successive passages on nutritious agar medium were performed, followed by incubation for 24 h, at 37 °C.

The qualitative screening of the antimicrobial properties was performed by an adapted spot diffusion method, in accordance with CLSI standard (Clinical Laboratory Standard Institute, 2021) [44]. Microbial suspensions of 1.5 × 10^8^ CFU/mL (corresponding with 0.5 McFarland standard density) and 6 × 10^8^ CFU/mL (corresponding with 2 McFarland standard density) were used in the experiments. Microbial inoculums were seeded on the specific agar medium (Muller Hinton agar for bacterial strains and Sabouraud agar for yeast strain) and an amount of approximatively 10 µL of each sample was spotted over: TiO_2_ anatase pigment doped with 2% copper metal (100 mg/mL in distillate water) (sample 1), photocatalytic washable paint ALINNA (sample 2), and normal washable paint without pigment used as control (sample 3). The plates were left at room temperature to ensure the equal diffusion of the compound in the medium and then incubated at 37 °C for 24 h in two different conditions: exposure to blue light (470 nm) generated by a commercial light LED source with 470 nm wavelength emission spectrum (Appendix A) and exposure to visible light.

The sensibility of tested microbial strains was evaluated by measuring the diameters of the inhibition zones that appeared around the spot. To avoid the variations between sample droplet diameters, the results were expressed as the ratio of droplet diameter to total diameter (droplet diameter + area of inhibition). The lower the ratio, the higher the antimicrobial effect.

For the quantitative evaluation of the antimicrobial efficiency, an adapted standard method was used, corresponding with ISO 22196/2011 standard [45]. Quantitative testing was performed in 6-well plates and 1 mL of each sample was spread in the wells in order to form a continuous layer on the bottom of the well. The plates were maintained for 24 h for drying and the first set (set 1) was inoculated with 500 µL of the two different microbial suspensions prepared in 1/500 diluted nutrient broth (distillate water/nutrient broth = 1/500), with final density 1.5 × 10^6^ CFU/mL and 6 × 10^6^ CFU/mL. Set 1 was incubated at 37 °C for 2 h in the wet chamber, in the same two differing conditions: blue light (470 nm) and visible light (considered T2h). 

A second set (set 2) of 6-well plates covered with tested samples and inoculated with 500 µL of the two different microbial suspensions was used as microbial cell recovery control, with no incubation time (considered T0h).

In the following steps, both sets (T2h and T0h) of tests were performed similarly. Thus, 1 mL of neutralizing agent represented by 1% sodium thiosulphate was added to the samples and quantitative evaluation using viable cell count method (VCC method) was performed. 

### 2.10. Statistical Analysis

Biological results were analyzed using one-way ANOVA repeated measures test. All statistical analyses were performed using GraphPad Prism Software, v. 5.03 428 (GraphPad Software, La Jolla, CA, USA, www.graphpad.com, accessed on 20 May 2020). Significance difference was noted as: * for *p* < 0.05, and ** for *p* < 0.01.

### 2.11. International Certification

The antimicrobial photocatalytic activity of the new photocatalytic antibacterial paint ALINNA was quantitatively and qualitatively evaluated in the international laboratory and certified based on the following standards: EN 14885: 2015—Chemical disinfectants and antiseptics and ISO 27447: 2009 Standard—Test methods for antibacterial activity of semiconducting photocatalytic materials (see Appendix A).

## 3. Results

The purpose of this paper is to validate a new concept of light-activated photocatalytic antimicrobial paint (ALINNA) in the visible spectrum which is based on international patent [36]. The new paint is produced based on resins containing photocatalytic pigment based on copper-doped TiO_2_ anatase. The experiments aimed to evaluate the photocatalytic activity in the visible spectral domain, the toxicity of new dye compositions, but also the antibacterial activity in visible light. Since it is desired to introduce the product to manufacture, it was intended that all the experimental and validation methods applied to study the photocatalytic antibacterial properties were to be performed according to the standardized certification requirements of the industry.

### 3.1. Evaluation of the Physical Properties of the Photocatalytic Antibacterial Washable Paint

Batches of copper-doped TiO_2_ pigment were manufactured in the small production laboratory of a paint factory, based on the method described in Romanian patent application [36]. The pigment obtained was used to produce and industrially homologate a washable paint with photocatalytic antibacterial properties, using a production recipe described in literature [37].

All the physical parameters obtained for the new washable biocidal paint with photocatalytic properties (ALINNA) fall within the limits of the industrial imposed standards: the density was determined by the pycnometer method and an average value of 1.61 g/cm^3^ was obtained (SR EN ISO—2811-1-2002); Brookfield viscosity was determined at the value B = 14,600 (in accordance with ISO 2884-2: 2003). Moreover, the test was performed to determine the drying time (in accordance with ISO 1517/1999). A touch drying time of 30 min at 20 °C was obtained and total drying in 2 h at 20 °C (results are within the limits of EN ISO 1517/1999).

The washability of the new paint was tested by determining the wet abrasion resistance (in accordance with EN ISO 11998: 2007). A mass loss per unit area of 117.63 g/m^2^ was obtained. There was a loss of 59 μm film thickness. The results obtained were within the limits required by the standard EN ISO 11998: 2007 for class 3 wet abrasion resistance.

### 3.2. UV-Vis Optical Absorption Spectra

To study the optical properties and light absorption effects, UV/VIS optical absorption spectra have been collected for the copper-doped TiO_2_ anatase pigment and for the photocatalytic antimicrobial paint in which the copper-doped TiO_2_ anatase pigment has been introduced.

Figure 4 shows the UV-Vis optical absorption spectra in the 330–1000 nm wavelength range for copper-doped TiO_2_ anatase pigment compared to that of non-doped TiO_2_ anatase pigment sample. A strong absorption can be observed in the range 330–370 nm, relatively equal for both samples. For the copper-doped TiO_2_ anatase pigment (2%), a strong increase in light absorption in the wavelength range between 400 and 700 nm, with a wide maximum between 420 and 620 nm, has been observed.

UV-Vis optical absorption tests have been performed on paint samples in which different concentrations of copper-doped TiO_2_ anatase had been introduced. Figure 5 compares the UV-Vis optical absorption spectra for paints containing concentrations of 0%, 2.25%, 4.5%, and 9% copper-doped TiO_2_.

The introduction of copper-doped TiO_2_ pigment into the paint led to a different light-absorption profile of the paint film. It can be noticed that the intensity of the absorption, which can be extrapolated from the reflection curve, varies depending on concentration. However, for concentrations higher than 4.5%, this absorption is relatively stationary. For example, the reflectance curve upon light absorption for paint with a concentration of 4.5% is almost similar to that of 9% concentration.

### 3.3. X-ray Powder Diffraction (PXRD) Characterization

X-ray diffraction has been used to determine the composition of the powder. The XRD pattern shows sharp diffraction peaks which indicate the crystallinity of the system (Figure 6). Copper dopant was observed in cubic phase (PDF Card No. 01-080-5762) and titanium dioxide was found to be in the form of tetragonal anatase phase (PDF Card No. 00-064-0863).

### 3.4. Scanning Electron Microscopy

The SEM results show agglomerated spherical TiO_2_ crystallite particles with an average diameter ranging between 100 and 200 nm (Figure 7). Copper plates can be observed in the center of the image.

The samples have also been analyzed using EDX (Figure 8 and Table 2). From the presented data, the elemental composition of the powder (atomic percentage) was determined (Table 2). The results demonstrate the presence of Cu deposits on TiO_2_ that can be seen in spectrum 8, 9, and 10 (Table 2), confirming the SEM and XRD results.

### 3.5. X-ray Photoelectron Spectroscopy

The X-ray photoelectron spectroscopy (XPS) analysis of the sample was conducted by using a monochromatized Al Kα X-ray source (1486.7 eV), with the photoelectrons recorded by a hemispherical (180 mm radius) electron energy analyzer with a reference intensity of over 50.000 counts per seconds (cps) and maximum spectral resolution of 0.7 eV. The X-ray source operated at 300 W (15 kV × 20 mA), with spot size of 0.7 mm. Partial charge compensation was reached by using a flood gun operating at 1.52 A filament current, 2.73 V charge balance, and 1.02 V filament bias. The base pressure during the measurements was around 10^−8^ Pa.

The XPS investigation was carried out to provide evidence for the nature of bonds and to discover the Cu oxidation state in the copper-doped TiO_2_ anatase pigment. The core level spectra (Figure 9) have been deconvoluted using Voigt functions (Lorentzian and Gaussian widths), based on the methods described in reference [46] and are presented in Figure 9 for: (a) O 1s levels; (b) for Ti 2p levels; and (c) for Cu 2p levels. All the spectra are deconvoluted with a minimum number of components in order to obtain a reasonable fit.

The atomic composition, Table 3, has been determined by using the integral areas provided by the deconvolution procedure normalized at the atomic sensitivity factors [47], taking into consideration a slight contamination of the surfaces with S 2p, Na 1s (residues from the manufacturing process) and C 1s. The C 1s core level was assigned to the adventitious carbon on the sample surface that is nearly unavoidable, and it was not taken into consideration for the atomic composition. The binding energy scale for all XPS spectra was calibrated to the C 1 s standard value of 284.6 eV. The atomic concentration ratios are presented in Table 3 and were computed from the sum of integral intensities by use of XPS atomic sensitivity factors. 

In the case of O 1s we can observe in Figure 9a the presence of three components: C1 (530.255 eV) can be associated with Cu_2_O [48]; C3 (529.143 eV) corresponds to the oxygen atoms in TiO_2_ phase; and C2 (532.59 eV) corresponds to the surface contamination with carbon also found in the XPS spectrum. Figure 9b shows the two peaks corresponding to the core level binding energies of Ti 2p (1/2 and 3/2) fitted with two components; the separation between the doublet (spin-orbit split) of 5.76 eV is characteristic of Ti^4+^ oxidation state in TiO_2_ [49]. Both components, C1 (457.917 eV) and C2 (458.441 eV) indicate the presence of the two phases of titanium dioxide (TiO_2_). Figure 9c shows the presence of Cu in the sample and the existence of Cu^1+^ and Cu^2+^ was confirmed by XPS. The value found in literature [50] (932.4 eV) for the Cu2O (Cu I) is very similar to the binding energy of C2 (932.394 eV). The other component, C1 (931.439 eV) can be associated with CuO (Cu II). The relative concentration ratio of Cu^1+^ to Cu^2+^ phases was found to be 0.95.

### 3.6. Specific Migration Tests of Heavy Metals

The result of the heavy metals’ (Pb, Cd, and Cu) specific migration determined after 10 days’ extraction at 20 °C in simulant B (acetic acid 3% solution) using ICP-MS technique are presented in Table 4.

### 3.7. Photocatalytic Activity Analysis

The estimation of the photocatalytic activity was conducted by measuring, at different intervals of time, the absorbance of a model contaminant, respectively, an aqueous solution of methylene blue (MB) 20 mg/L, using different light sources. The methodology adopted during the experiments allowed us to make the difference between the total photocatalytic activity and photocatalytic activity in ultraviolet and visible range, respectively. 

#### 3.7.1. Photocatalytic Activity during Irradiation with Visible Light (400–800 nm)

The evaluation of the antimicrobial paint’s photocatalytic behavior after irradiation with visible light is presented in Figure 10. The discoloration of the MB solution in the lighting conditions specified above and in the presence of ALINNA photocatalytic paint, takes place in proportion of 96%, after 3360 min. 

However, the absorbance of MB solution decreases linearly up to a discoloration degree of 75%, which corresponds to only 1800 min of exposure (Figure 10b). Afterwards, the process continues nonlinearly, probably due to the MB desorption-adsorption-diffusion equilibria at the interface solution-paint that controls the last part of the photocatalytic process, up to a discoloration of 96%.

In order to estimate the photocatalytic activity correctly, from variations of the MB absorbance, only the linear domain was used. Due to the processes that take place simultaneously, there is the possibility of overestimating the photocatalytic activity in the visible field, most probably because of the superimposing of photochemical decomposition of MB during irradiation with visible light. 

The total specific degradation ratio during irradiation with visible light, in the presence of the photocatalytic paint, was R_tot_ = 1.57 × 10^−5^ mol/m^2^h.

Figure 11 shows the time variation of the MB absorbance upon irradiation in visible light, in the absence of the photocatalytic paint. We determined that the discoloration of the aqueous MB solution is a slower process and that it takes place linearly. The linearity of the process is caused by a lack of associations between MB molecules in solution, at the concentration of 20 mg/L.

The total specific degradation ratio during irradiation with visible light, in the absence of the photocatalytic paint, was R_irr_ = 1.11 × 10^−5^ mol/m^2^h.

As we have previously affirmed, in order to avoid overestimation, the values of the absorbance are subtracted from those obtained in the case of the global process, because of the linearity.

As a result, the average specific photocatalytic activity upon illumination with visible light is P_MB_ = 0.46 × 10^−5^ mol/m^2^h and is achieved by the subtraction: P_MB_ = R_tot_ − R_irr_.

#### 3.7.2. Photocatalytic Activity during Irradiation with Near Ultraviolet Light (300–400 nm)

The evaluation of the photocatalytic paint behavior after irradiation with near ultraviolet light (UVA) is presented in Figure 12. The results show that the discoloration of the aqueous solution of MB takes place linearly and fast, up to about 45%, after only 270 min.

If we do not consider parallel processes which can take place during the irradiation with UVA light, the total specific degradation rate in ultraviolet determined by us is R_tot_ = 3.75 × 10^−5^ mol/m^2^h.

However, when irradiating the MB solution with UVA radiation, in the absence of photocatalytic antibacterial paint, the results (Figure 13) show that a process of discoloration of the aqueous solution of MB takes place almost linearly at a very low rate. 

In this case, the specific degradation rate in the absence of the photocatalytic paint is R_irr_ = 0.6 × 10^−5^ mol/m^2^h.

The average of specific photocatalytic activity under illumination of the photocatalytic antimicrobial paint with UVA light was P_MB_ = 3.15 × 10^−5^ mol/m^2^h.

#### 3.7.3. Photocatalytic Activity during Irradiation with Arc-Xenon Light (300–800 nm)

The behavior of the photocatalytic antimicrobial paint on irradiation with arc-xenon light, which is currently used to simulate sunlight, has also been assessed. The discoloration of the MB solution (Figure 14), in the presence of the photocatalytic antibacterial paint, takes place linearly on the considered range, up to about 60%, in only 180 min.

The total specific degradation rate determined during irradiation was R_tot_ = 2.95 × 10^−5^ mol/m^2^h.

In the absence of photocatalytic paint, the discoloration of the aqueous MB solution is carried out linearly, as shown in Figure 15. The specific degradation rate determined during irradiation of the MB aqueous solution was R_irr_ = 2.31 × 10^−5^ mol/m^2^h.

The average specific photocatalytic activity of photocatalytic antimicrobial paint, upon illumination with simulated sunlight (arc-xenon) light, was P_MB_ = 0.64 × 10^−5^ mol/m^2^h.

### 3.8. Antimicrobial Tests

The qualitative screening of the antimicrobial activity evaluated the efficiency of the three samples in two different conditions of incubation by measuring the diameters of the inhibition zone expressed by each tested microbial strain and finally, by expressing the results as diameter ratio. The lower the ratio, the higher the antimicrobial effect. 

The recorded values for diameter ratio demonstrated statistically significant differences for *S. aureus* ATCC 25922 inhibition zone after incubation in blue light for 2 h, with *p*(T ≤ t) = 0.05 for McFarland 0.5 suspension and *p*(T ≤ t) = 0.02 for McFarland 2 suspension, compared with visible light conditions. This result demonstrates that the photocatalytic activation of 2% copper-doped TiO_2_ pigment and photocatalytic paint ALINNA increases the efficiency of the antibacterial activity for these two samples, even when the bacterial cell density is higher (Figure 16). 

For the other tested microbial strains, *P. aeruginosa* ATCC 27853 and *C. albicans* ATCC 10231, although the data recorded in the two graphs (Appendix A) showed differences between the diameters of the inhibition zones after incubation in the dark and blue light (470 nm), they were not statistically significant (*p* > 0.05). However, a potentiation of the antimicrobial activity of washable paint ALINNA can be observed after exposure of the McFarland 0.5 microbial suspension (corresponding to 1.5 × 10^8^ CFU/mL) on blue light (470 nm) for 24 h, the diameters ratio of the inhibition zone varying from 0.66 (in visible light) to 0.5 (in blue light) for the *P. aeruginosa* strain (Appendix A), or from 0.5 (in visible light) to 0.3 (in blue light) for the *C. albicans* strain (Appendix A). When the microbial cell density was higher (6 × 10^8^ CFU/mL), the diameters of the inhibition zone has not significantly changed after 24 h hours’ exposure in blue light.

In order to demonstrate the photoactivation reaction of the 2% copper-doped TiO_2_ pigment and to quantify the antimicrobial activity of the tested products following the photoactivation, viable cell count method (VCC method) was performed (in accordance with standard method ISO 22196/2011) using two test conditions (visible light and blue light—470 nm) and two cell suspensions with different final densities (1.5 × 10^6^ CFU/mL and 6 × 10^6^ CFU/mL). 

In our experiment, the two tested products (TiO_2_ anatase 2% copper-doped pigment and washable paint ALINNA) significantly inhibited the multiplication of all tested strains under both incubation conditions (the *P(T ≤ t) < 0.05) (Figure 17a,b, Figure 18a,b, and Figure 19a,b). The CFU/mL values were equally low for both suspensions with different cell densities (1.5 × 10^6^ CFU/mL and 6 × 10^6^ CFU/mL), demonstrating the very good efficiency of the products after 2 h of contact, even when the microbial load is higher.

The preservation of the inhibitory effect of the two products by exposure to blue light supports the hypothesis of photoactivation mechanism of the copper-doped TiO_2_ pigment which triggers the biochemical processes with toxic effect on microbial cells.

## 4. Discussion

The production start of the new type of photocatalytic antimicrobial paint requires thorough testing for product evaluation and characterization, in accordance with the approval standards required by ISO 18451-1: 2019 [51], as well as with the certification provisions of paints. After obtaining the batch of photocatalytic antimicrobial paint ALINNA, the manufacturer performed tests in the quality control laboratory to evaluate the product. The obtained results are framed within the standards of washable paints for industrial certification of the product. Tests have shown that a product is obtained in the form of a paint suspension, perfectly reproducible from the perspective of physical properties, which forms a robust, perfectly adherent film on the concrete surfaces and has the property of uniformly covering the applied surfaces, this being an essential feature for paints used for decorative and surface protection.

Interpretation of the UV-Vis diffuse reflectance spectrum of 2% copper-doped anatase TiO_2_ pigment, which was compared with the pure anatase TiO_2_ pigment used in the doping process, provides particularly valuable information on the photocatalytic process generated by these pigments. In the case of both pigments, a specific photocatalytic response was obtained in the 350–390 nm spectral area.

Unlike pure anatase TiO_2_ pigment that does not absorb light at all in the visible spectrum, for the 2% copper-doped TiO_2_ pigment, an optical absorption curve over the entire visible spectrum range of 400–700 nm was obtained, whose percentages are high and relatively constant over the range, with a higher level in the 420–670 nm domain. Thus, it was demonstrated that this pigment strongly absorbs visible light. 

Obtaining a relatively constant absorption level over the entire visible spectrum indicates that the intensity of the photocatalytic process is constant, regardless of the type of light (artificial or natural) used to initiate the photocatalytic reaction. Therefore, the photocatalytic antibacterial product can be used with the same results both in spaces illuminated with natural light and also in enclosed spaces that are artificially lit with lighting fixtures.

Another important aspect resulting from the interpretation of the absorption spectra is revealed in the intensity of the photocatalytic process. The fact that this relatively uniform level of visible light absorption has values ranging from 0.5 to 0.45 demonstrates high efficiency in generating photocatalytic reactions on the surface of the copper-doped TiO_2_ pigment across the visible spectrum (400–700 nm).

After obtaining the copper-doped TiO_2_ pigment, tests have been performed to introduce the pigment into a paint composition. Generally, most paint formulas contain between 5 wt% and 15 wt% TiO_2_. In order to evaluate the optimal amount of TiO_2_ to be introduced into the paint, different samples of paint were prepared in which 2.25, 4.5, and 9 wt% of copper-doped TiO_2_ (2% Cu) were introduced and absorption spectra have been collected. The optimum pigment concentration of copper-doped TiO_2_ in the paint was determined to be 4.5%. Although we would have expected the light absorption to increase above this value, and therefore the photocatalytic activity also to increase, it has been observed that the paint absorption of 4.5 wt% is almost identical to that of 9 wt%. This is attributed to the phenomena of diffusion and dispersion and is a future path of research for these paints.

The density functional theory (DFT) calculations were performed in order to characterize the electronic and structural properties of the TiO_2_ pigment doped with copper and to understand the mechanism of photocatalytic activation of copper-doped TiO_2_ pigment. The quantum solid state code CASTEP [52] was used for the DFT calculations, with a high-accuracy spin-polarized calculation scheme (plane-wave ultrasoft pseudopotential, exchange-correlation functional PBE, energy cutoff of 325 eV, Brillouin zone—discretized by Monkhorst-Pack scheme, with a spacing of 0.04 Å^−1^). The quality of the used calculation scheme is validated by the confirmation of the experimental lattice parameters for anatase and rutile TiO_2_ and copper. The structural model of the copper-doped TiO_2_ pigment is given in Figure 20.

The charge density difference was calculated as:Δρ = ρ_Cu(001)/TiO2(001)_ − (ρ_Cu(001)_ + ρ_TiO2(001)_)
where ρ_Cu(001)/TiO2(001)_ is the electron density of the total Cu(001)/TiO_2_(001) system, and ρ_Cu(001)_ and ρ_TiO2(001)_ are the unperturbed electron densities of the two surfaces, Cu(001) and TiO_2_(001), respectively. 

The charge density difference is localized in the interface area TiO_2_(001) and Cu(001) where the Cu atoms form covalent chemical bonds with the oxygen atoms from the first surface layer of the surface TiO_2_(001).

It is interesting that the SEM images also demonstrate the formation of agglomerates representing copper plates in metallic form, on the surface of the TiO_2_ pigment, which would justify the model obtained by computational simulation (Figure 20). In order to better understand how the photoexcitation of the doped pigment is conducted, but also how the electrons are delocalized, further investigations are necessary.

The photocatalytic activity of the paint was determined using an internal method developed based on the standard DIN 52980: 2008-10 [53], respectively, ISO 10678—2010 [54]. This method was chosen because it is suitable for the evaluation and certification of industrial photocatalytic products. The results demonstrated that the new photocatalytic antimicrobial paint showed photocatalytic activity both when irradiated exclusively with light from the near UV domain and when irradiated exclusively with light from the visible field. The average specific photocatalytic activity when illuminated with artificial light in the visible range (400–800 nm) is equal to the average specific photocatalytic activity when illuminated with xenon arc light (simulating natural light), a predictable result from spectral curve measurements of copper-doped anatase TiO_2_ pigment. Interestingly, the difference between the average specific photocatalytic activity when illuminated exclusively with UV-A, whose value is P_MB_ = 3.15 × 10^−5^ mol/m^2^h, differs only by an order of magnitude compared to the average specific photocatalytic activity when illuminated exclusively with visible light that has a value of P_MB_ = 0.46 × 10^−5^ mol/m^2^h.

The toxicity of the photocatalytic antimicrobial paint was evaluated by the ion migration test. Although the body doping percentage was quite high, the test did not show copper ions in free migration. This fact can be correlated with the SEM image where it is observed that copper forms agglomerations of metal particles closely adhering to the surface of the TiO_2_ crystal, a fact also observed in the modeling made on the CASTEP program. The lack of copper ions during migration leads us to the idea that through the doping process the copper atoms form coordinative bonds with oxygen atoms on the surface of the TiO_2_ crystal, and the antimicrobial effect of this compound is due only to photocatalytic reactions. The ion migration test also certifies that we have a non-toxic, safe product for humans.

Numerous articles have investigated the action of these semiconductor metal oxides; the photocatalytic mechanisms have been studied, as well as how these mechanisms irreversibly affect the pathogens’ viability [55,56,57,58]. Walker at al. (2017) demonstrated, for the first time, the effectiveness of the light-activated antimicrobial surface against yeast, viruses, filamentous fungi, and fungus-like organisms [59]. Research results regarding the inactivation of *Escherichia coli* by photocatalysis involving TiO_2_ nanoparticles alone or in transparent coatings (varnishes) were recently published [60,61]. The antibacterial activity of TiO_2_ was evaluated through two types of experiments under UV irradiation: (i) in slurry with physiological water (stirred suspension); and (ii) in a drop deposited on a glass plate. The results confirmed the difference in antibacterial activity between simple drop-deposited inoculum and inoculum spread under a plastic film, which increased the probability of contact between TiO_2_ and bacteria (forced contact). Experiments were also carried out at the surface of transparent coatings formulated using nanoparticles of TiO_2_. The results showed significant antibacterial activities after 2 h and 4 h and suggested that improving the formulation would increase efficiency [25].

Guffey and Wilborn (2006) [62] studied the effect of radiation with a wavelength of 405 nm and 470 nm on the species *Staphylococcus aureus*, *Pseudomonas aeruginosa,* and *Propionibacterium acnes*; they noted that both types of lighting showed bactericidal effect on *P. aeruginosa* and *S. aureus*, but not on *Propionibacterium acnes*. Different scientists studied the effect of blue light (470 nm) on several strains of methicillin-resistant *Staphylococcus aureus* and the results showed that the number of colony-forming units (CFUs) of the samples exposed to light radiation of 470 nm was smaller than in the case of non-irradiated ones [63,64]. Maclean et al. (2009) [52] studied the effect of light with a wavelength of 405 nm on several microbial strains responsible for nosocomial infections and observed that both Gram-positive and Gram-negative strains were inactivated after exposure. 

In the present study, the antimicrobial qualitative results demonstrate the photocatalytic activation of copper-doped TiO_2_ pigment and photocatalytic paint ALINNA in both conditions (blue light and visible light 470 nm), the ROS molecules generated by the products causing inhibitory effects against all tested microbial strains (*S. aureus* ATCC 25922, *P. aeruginosa* ATCC 27853, and *C. albicans* ATCC 10231), even when the microbial cell density was higher. In order to confirm the photoactivation reaction of the copper-doped TiO_2_ pigment and ROS molecules release, and to quantify the antimicrobial activity, the viable cell count method (VCC method) was performed, using two test conditions (visible light and blue light—470 nm) and two cell suspensions with different final densities (1.5 × 10^6^ CFU/mL and 6 × 10^6^ CFU/mL). The products confirmed the qualitative results, showing microbicidal activity after 2 h of exposure of tested strains in both concentrations, with significant decrease of CFU/mL values compared with control growth controls and paint control product. These results, together with the physicochemical characterization of tested products, entitle us to support with certainty the photocatalytic activation of copper-doped TiO_2_ product, and also, its efficiency as a new method of photocatalytic protection of medical supplies surfaces.

## 5. Conclusions

In this study, a multidisciplinary team of chemists, microbiologists, and physicists aimed to characterize a new photocatalytic-activated paint with antimicrobial efficiency to be certified and produced on an industrial scale. The new photocatalytic antimicrobial method uses a type of photocatalytic paint that is active in the visible spectral range. Tests have shown that the new product forms a robust, perfectly adherent film on the concrete surfaces, uniformly covering the applied surfaces, this being an essential feature for paints used for decorative and surface protection. The absorption spectra demonstrated a high efficiency in generating photocatalytic reactions on the surface of the copper-doped TiO_2_ pigment across the visible spectrum (400–700 nm). The new paint generated reactive oxygen species with inhibitory effect against all tested microbial strains after 2 h of exposure in visible light and blue light (470 nm) conditions, even when the microbial load was higher. These scientific observations demonstrated that the coating paint model is ecological and proves to be promising for reducing the microbial load in medical supplies surfaces and also for having no negative effects on the environment.

The obtained product was industrially certified and was used for the decorative coating and antimicrobial protection of the walls inside the palliative care health department of a hospital in Romania. Currently, long-term in situ studies are being carried out in that section, in order to observe the long-time activity of this paint and to evaluate the influence on the microbial load in the patient care areas of that health section.

## Figures and Tables

**Figure 1 materials-14-05307-f001:**
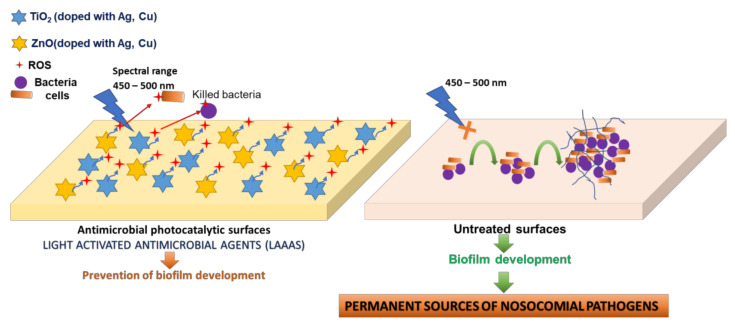
Proposed mechanism for the photocatalytic protection of medical supplies surfaces. The release of ROS molecules during the photoexcitation process prevents the irreversible adherence of the microbial cells to the treated surfaces and the generation of mature biofilms by altering the cell wall structures implicated in the adherence processes (left side of the figure). Instead, on untreated inert surfaces, microbial cells can adhere irreversibly and can generate mature biofilms, responsible for the spread of pathogenic strains, especially those with nosocomial properties (right side of the figure).

**Figure 2 materials-14-05307-f002:**
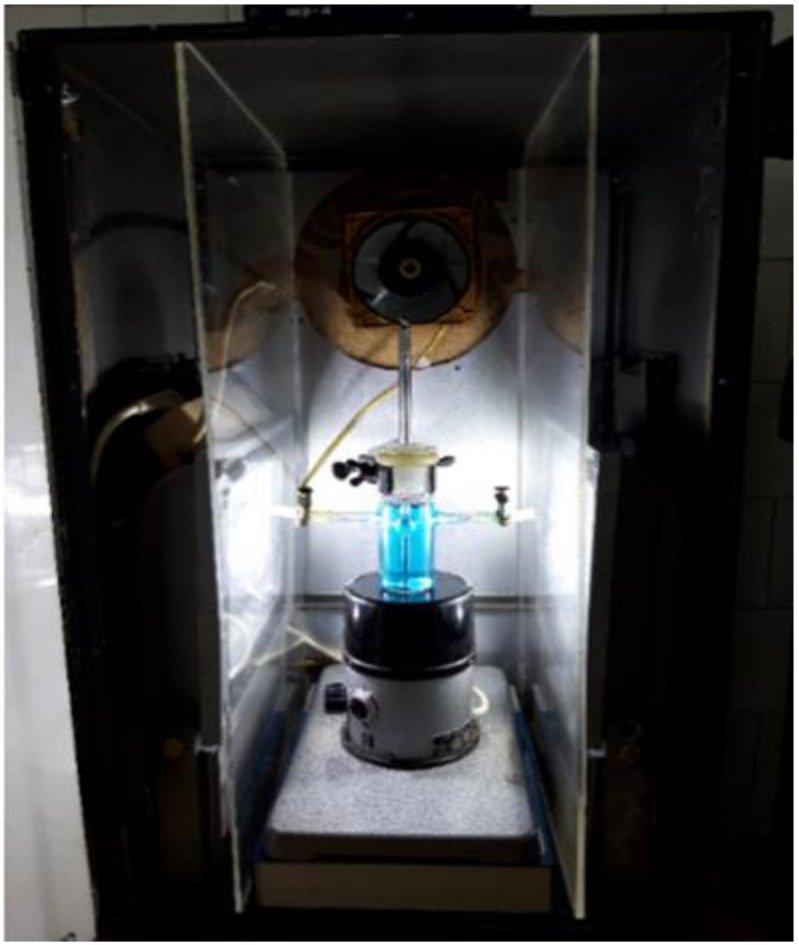
Original photocatalytic activity determination test chamber.

**Figure 3 materials-14-05307-f003:**
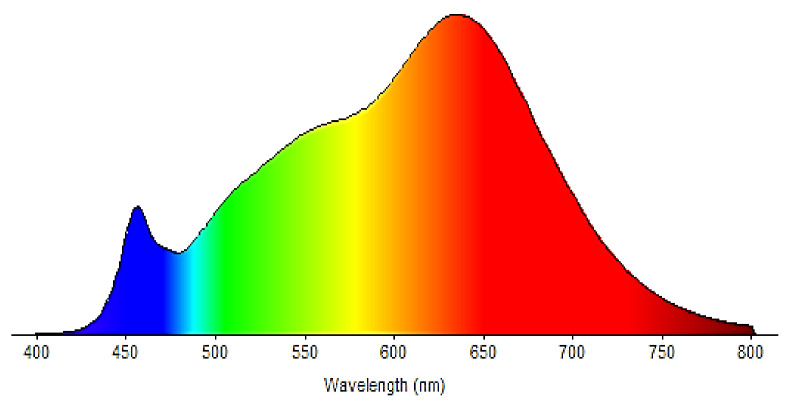
Emission spectrum of LED lamps (Stairville LED Power-Flood 100 W).

**Figure 4 materials-14-05307-f004:**
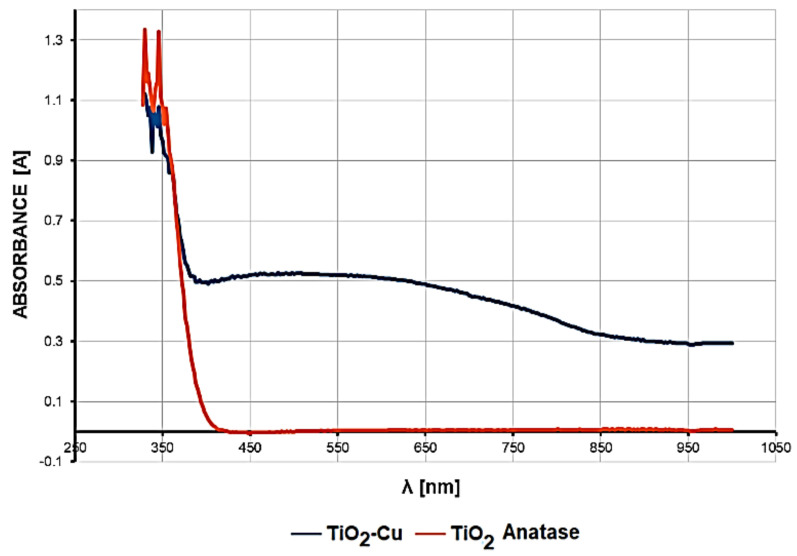
UV-Vis optical absorption spectra of 2% copper-doped TiO_2_ pigment compared to undoped TiO_2_.

**Figure 5 materials-14-05307-f005:**
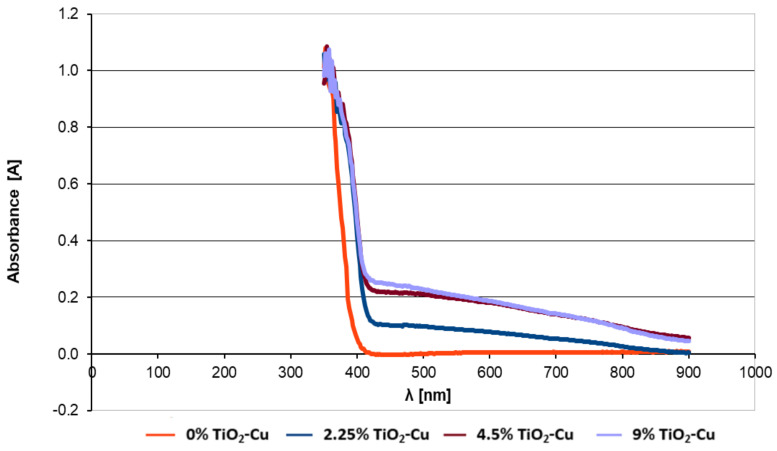
UV-Vis optical absorption spectra collected for paint containing different concentrations of copper-doped TiO_2_ pigment.

**Figure 6 materials-14-05307-f006:**
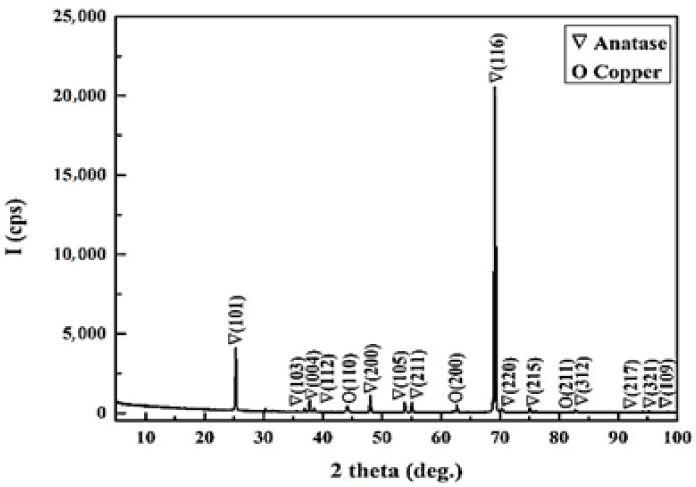
X-ray diffraction (XRD) pattern of TiO_2_ doped 2% Cu sample.

**Figure 7 materials-14-05307-f007:**
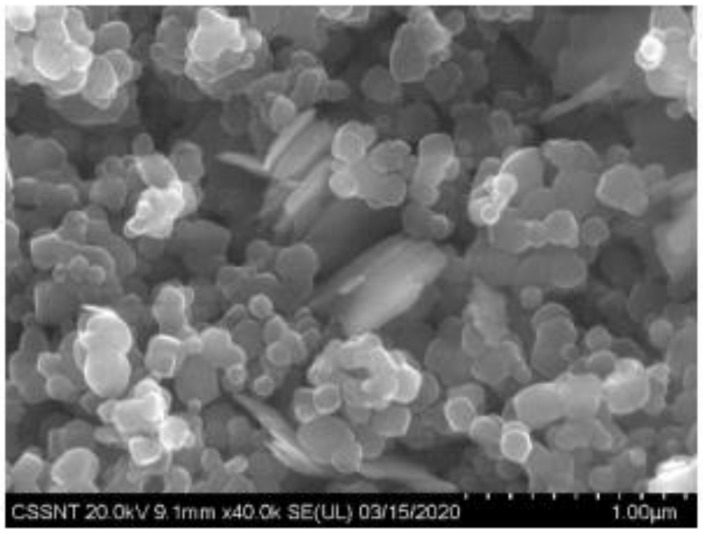
SEM micrographs of 2% copper-doped TiO_2_ pigment sample.

**Figure 8 materials-14-05307-f008:**
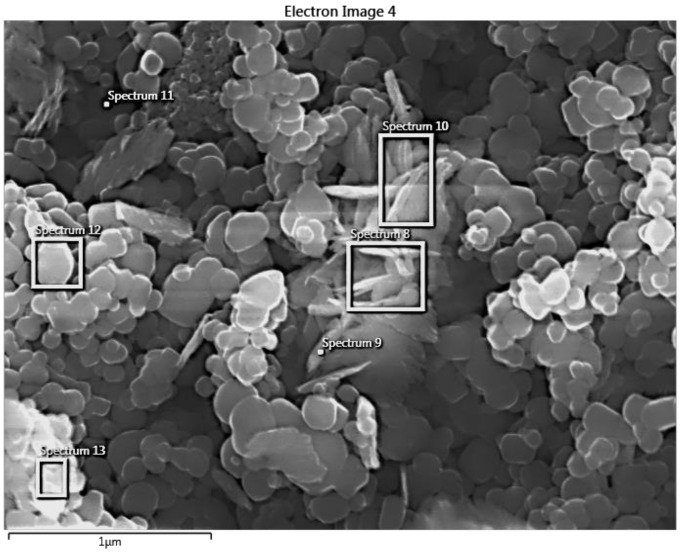
Determination of powder composition by EDX.

**Figure 9 materials-14-05307-f009:**
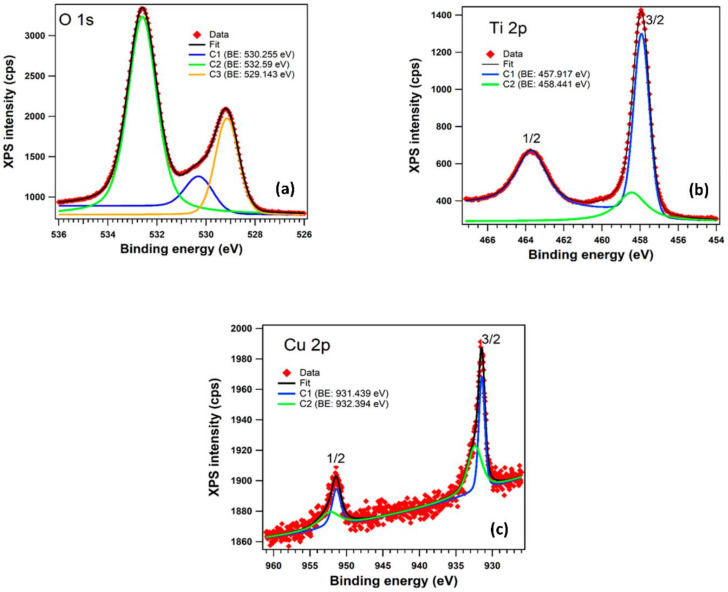
XPS spectra for core levels of 2% copper-doped TiO_2_ powder: (**a**) O 1s; (**b**) Ti 2p and (**c**) Cu 2p. In all graphs, the red symbols are the experimental data, black lines are the fits, blue and green lines are the individual component.

**Figure 10 materials-14-05307-f010:**
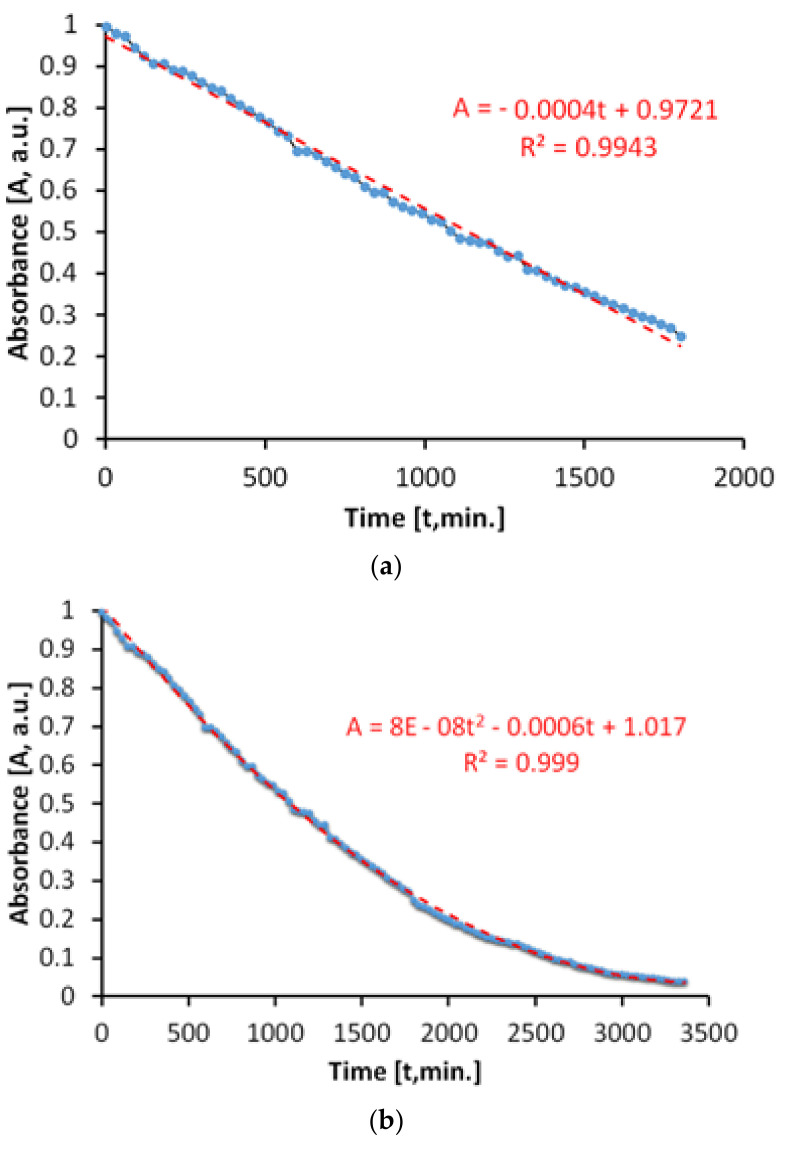
Time variation of MB absorbance upon irradiation exclusively with visible light in the presence of the photocatalytic paint; (**a**) linear domain; (**b**) until total discoloration.

**Figure 11 materials-14-05307-f011:**
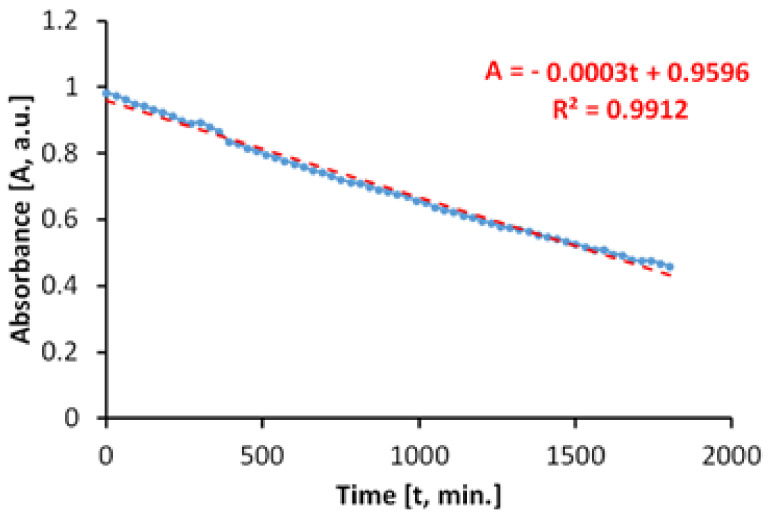
Time variation of MB absorbance during irradiation exclusively with visible light, in the absence of the photocatalytic paint.

**Figure 12 materials-14-05307-f012:**
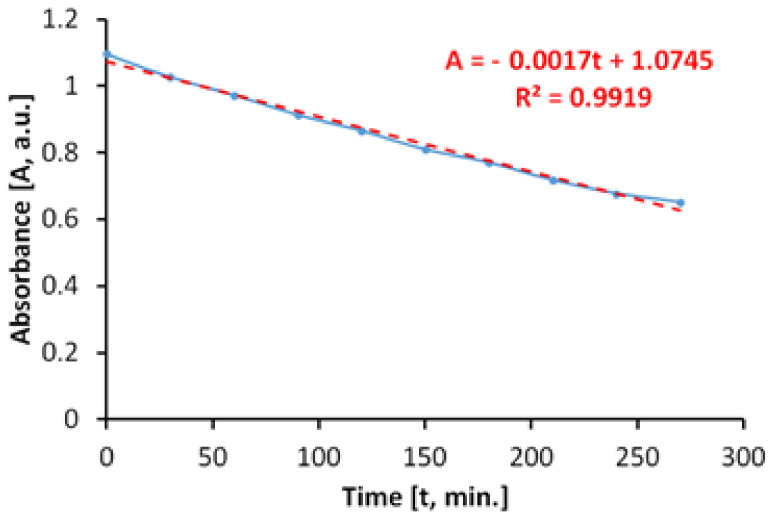
Time variation of MB absorbance when irradiating with light from the near ultraviolet range (300–400 nm) in the presence of the photocatalytic paint.

**Figure 13 materials-14-05307-f013:**
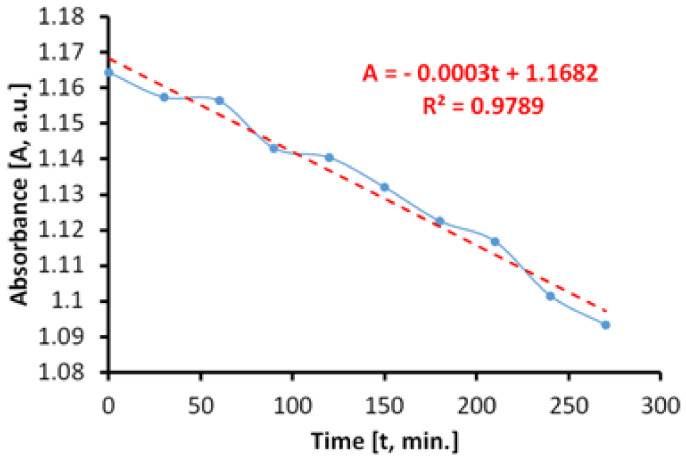
Time variation of MB solution absorbance by irradiating with light from the near ultraviolet range (300–400 nm) in the absence of the photocatalytic paint.

**Figure 14 materials-14-05307-f014:**
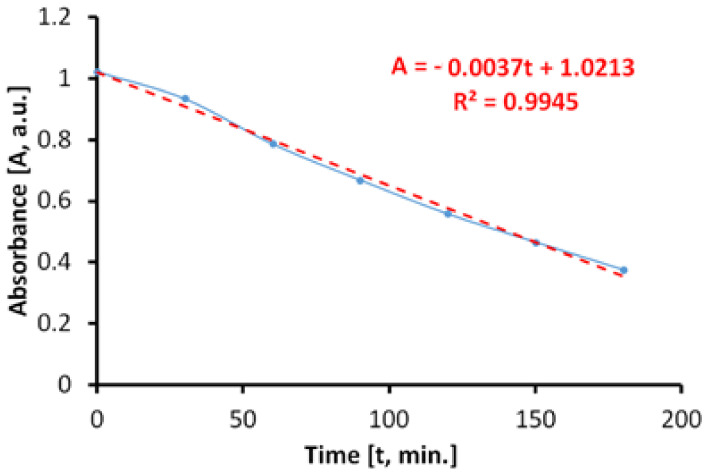
Time variation of MB solution absorbance by irradiating with arc-xenon light in the presence of the photocatalytic paint.

**Figure 15 materials-14-05307-f015:**
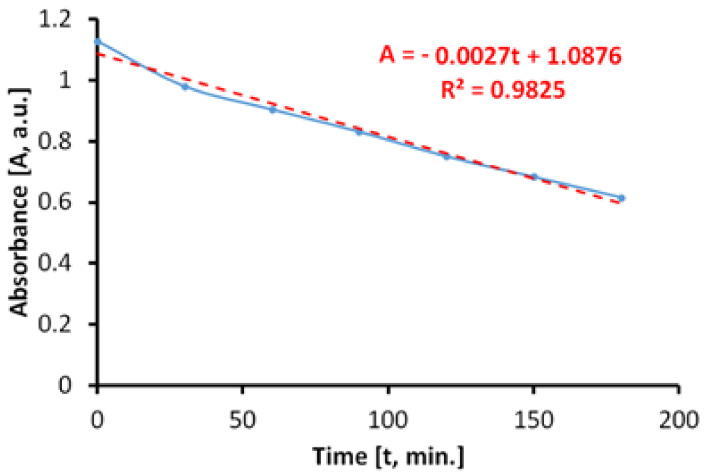
Time variation of MB solution absorbance by irradiating with arc-xenon light in the absence of the photocatalytic paint.

**Figure 16 materials-14-05307-f016:**
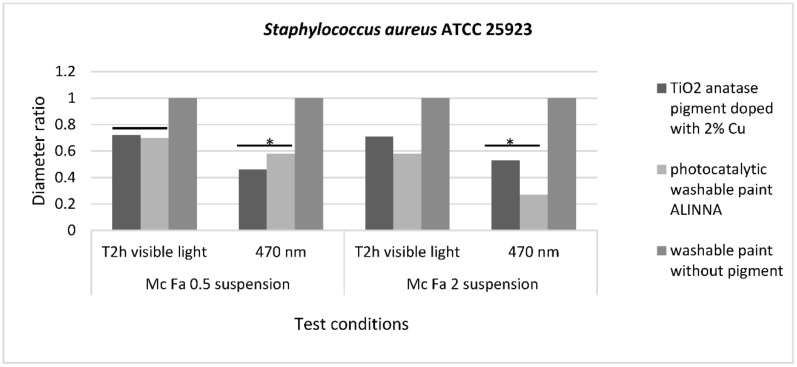
Graphic representation of inhibition zone diameters expressed as diameter ratio, obtained for *S. aureus* ATCC 25923 after incubation in two different conditions: visible light and blue light (470 nm). * 0.05 ≥ *p*(T ≤ t) > 0.001 is significant evidence of inhibitory effect manifested by 2% copper-doped TiO_2_ pigment on bacterial growth and multiplication.

**Figure 17 materials-14-05307-f017:**
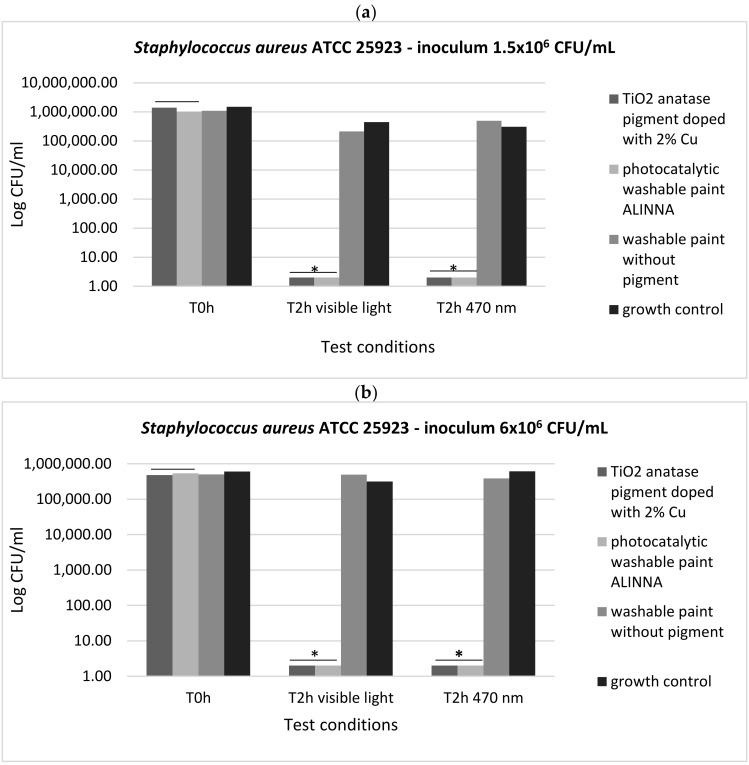
Graphical representation of the log10 values of colony-forming units (CFU)/mL representing the viable cells of *S. aureus* ATCC 25922 after the 2 h contact with tested products, in two conditions: blue light (470 nm) and visible light; (**a**) 1.5 × 10^6^ CFU/mL tested bacterial cell density; (**b**) 6 × 10^6^ CFU/mL tested bacterial cell density; * 0.05 ≥ *p*(T ≤ t) > 0.001 shows significant evidence of inhibitory effect manifested by doped pigment on bacterial growth and multiplication.

**Figure 18 materials-14-05307-f018:**
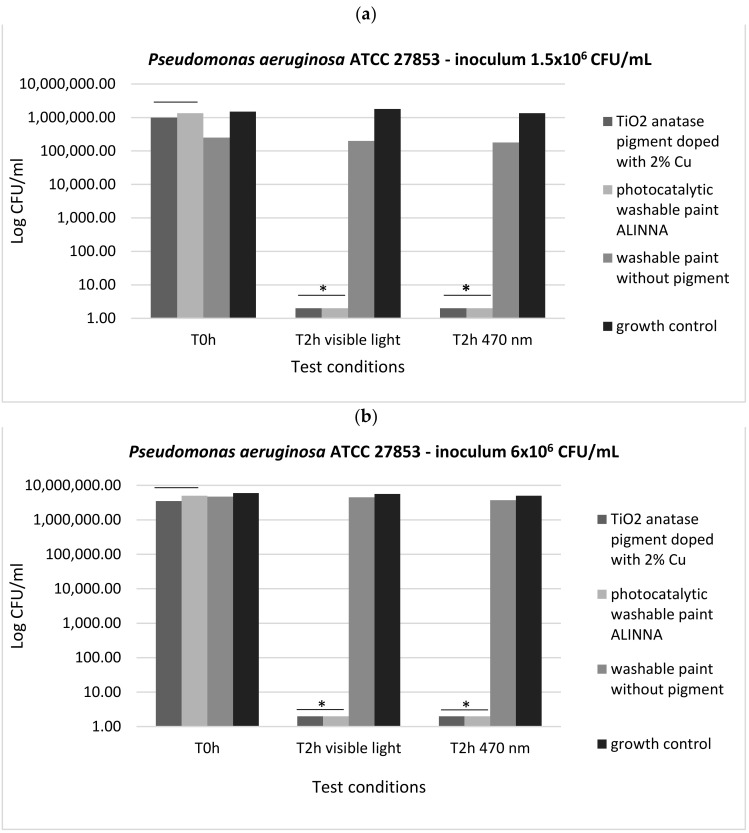
Graphical representation of the log10 values of colony-forming units (CFU)/mL representing the viable cells of *P. aeruginosa* ATCC 27853 after the 2 h contact with tested products, in two conditions: blue light (470 nm) and visible light; (**a**) 1.5 × 10^6^ CFU/mL tested bacterial cell density; (**b**) 6 × 10^6^ CFU/mL tested bacterial cell density; * 0.05 ≥ *p*(T ≤ t) > 0.001 shows significant evidence of inhibitory effect manifested by doped pigment on bacterial growth and multiplication.

**Figure 19 materials-14-05307-f019:**
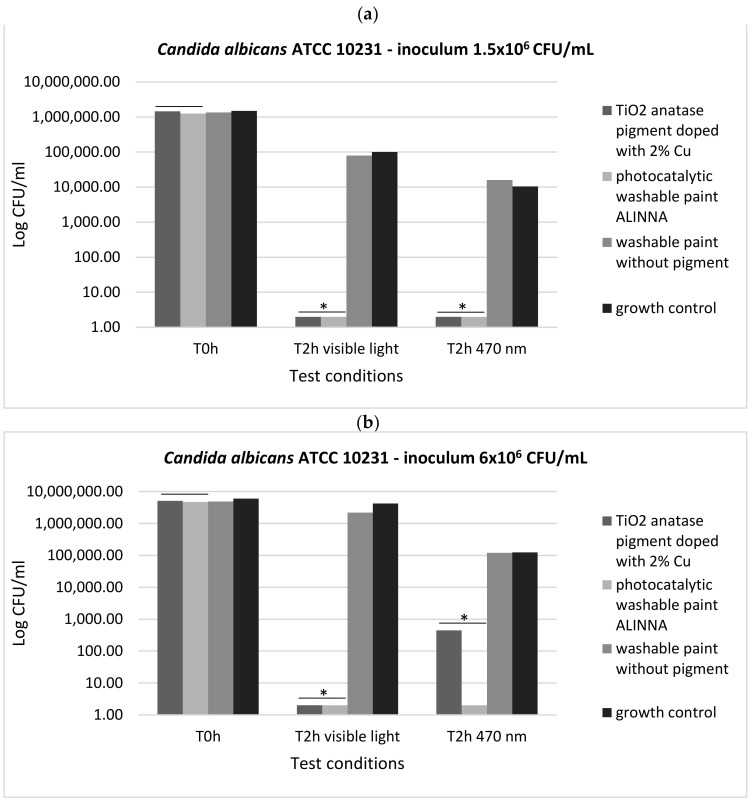
Graphical representation of the log10 values of colony-forming units (CFU)/mL representing the viable cells of *C. albicans* ATCC 10231 after the 2 h contact with tested products, in two conditions: blue light (470 nm) and visible light; (**a**) 1.5 × 10^6^ CFU/mL tested yeast cell density; (**b**) 6 × 10^6^ CFU/mL tested yeast cell density; * 0.05 ≥ *p*(T ≤ t) > 0.001 shows significant evidence of inhibitory effect manifested by doped pigment on bacterial growth and multiplication.

**Figure 20 materials-14-05307-f020:**
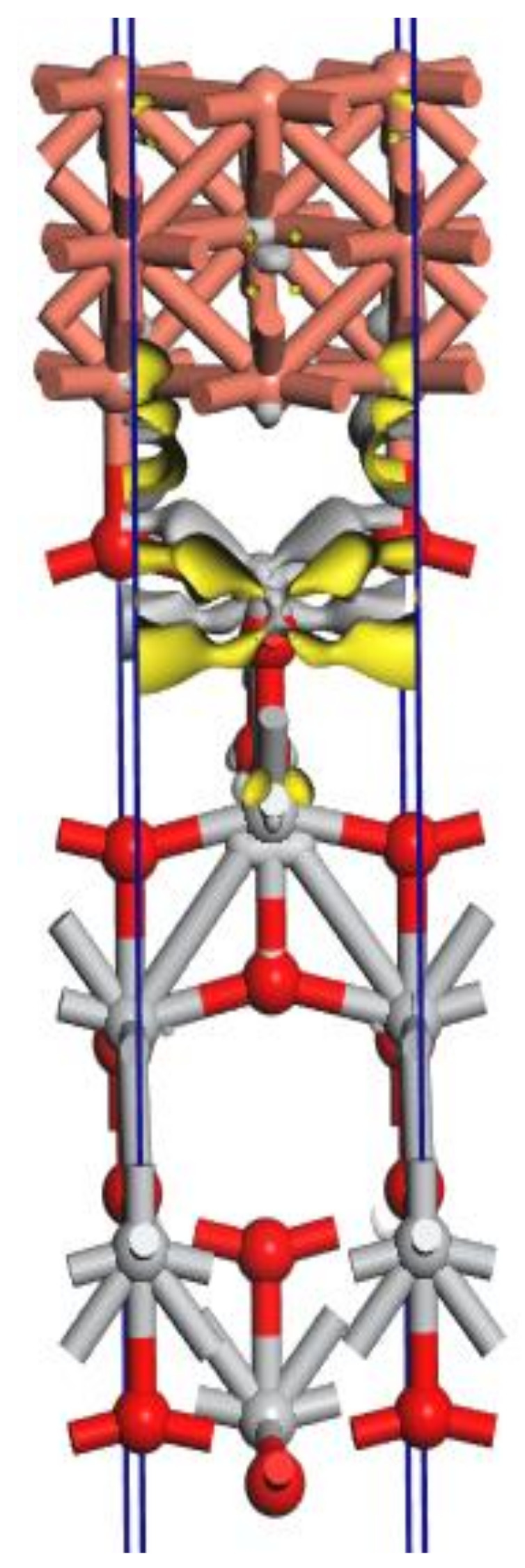
The charge density difference for the interface TiO_2_(001)-Cu(001) determined by DFT calculations (original design). It can be observed that copper (orange) forms coordinative bonds with oxygen at the surface of the TiO_2_ crystal (Ti—grey, O—red) and causes an electron displacement (yellow), which would suggest these properties of copper-doped TiO_2_.

**Table 1 materials-14-05307-t001:** NexION300Q instrumental parameters.

Parameter	Value
Cell gas	Argon
Nebulizer	Glass concentric
Spray chamber	Glass cyclonic
Nebulizer gas flow	0.89 L·min^−1^
Auxiliary gas flow	1.20 L·min^−1^
Plasma gas flow	16 L·min^−1^
RF power	1000 W

**Table 2 materials-14-05307-t002:** Composition of the powder.

Chemical Composition	Spectrum 8	Spectrum 9	Spectrum 10	Spectrum 11	Spectrum 12	Spectrum 13
O	74.94	75.24	75.07	66.46	82.36	79.65
Ti	17.22	18.07	18.87	33.18	17.20	19.34
Cu	7.84	6.69	6.06	0.35	0.44	1.01
Total	100.00	100.00	100.00	100.00	100.00	100.00
**Statistics**		**O**	**Ti**		**Cu**	
Max		82.36	33.18		7.84	
Min		66.46	17.20		0.35	
Average		75.62	20.64		3.73	
Standard Deviation		5.41	6.20		3.49	

**Table 3 materials-14-05307-t003:** Atomic composition computed from the sum of integral intensities.

	O	Ti	Cu	Other
Concentration	90.67%	7.487%	0.421%	1.4256%

**Table 4 materials-14-05307-t004:** Specific migration tests of heavy metals.

Specific Migration (µg/L)			Extraction Condition	
Pb	Cd *	Cu *	Time (Days)	Temperature (°C)
10.10	<1.0	<0.04	10	20 °C

Method detection limit for: Cd 1.0 µg/L; Cu 0.04 µg/L; * detection limit.

## Data Availability

The data underlying this article will be shared on reasonable request from the corresponding author.

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
