# Peer review of "Preliminary Study on Light-Activated Antimicrobial Agents as Photocatalytic Method for Protection of Surfaces with Increased Risk of Infections"

_materials, 2021, doi:10.3390/ma14185307_

Round 1
Reviewer 1 Report
The paper is well written, well structured with an in depth characterization of the photocatalytic system.
It's important for the reader to know how these results are different with respect to the conventional photocatalytic processes. In other words, which advantages can be found compared to what reported in literature?
Moreover, potential photocatalytic degradation scheme of the microbial strains during photocatalysis can be reported to understand potential by-products formed in the process.
Author Response
- The paper is well written, well structured with an in depth characterization of the photocatalytic system.
Respons: Thank you for your kindness and patience in reviewing our paper. We accept all the criticisms and observations that you have made on the paper and we thank you for supporting us in improving its scientific quality
- It's important for the reader to know how these results are different with respect to the conventional photocatalytic processes. In other words, which advantages can be found compared to what reported in literature?
Respons: The advantage of the method was emphasized by adding the explanatory paragraphs in the text.
- Moreover, potential photocatalytic degradation scheme of the microbial strains during photocatalysis can be reported to understand potential by-products formed in the process.
Respons: Thanks for the suggestion. I have added below the figure, an explanatory text to describe the process represented by the figure 1.

Reviewer 2 Report
It is an interesting paper written about photocatalytic coatings. The materials used here are not reported with much detail.
The introduction is well written. However, TiO2 materials as anti-bacterial photocatalysts have been studied for over 30 years. Thousands of publications have been written on the topic. Authors must clarify how this work is novel, and explain the limitations that have prevented TiO2 being used as a antibacterial photocatalyst in commercial applications so far.
Copper doped TiO2 has been often used in studies of photocatalytic materials. Authors should refer to the previous studies that have used Cu doped TiO2 films for the photocatalytic degradation of methylene blue in thin film form. How does copper affect the anatase to rutile phase transformation?
The authors have not reported the characteristics of their materials in a comparative manner. How were the TiO2 materials fabricated? Simply referring to a romanian patent is insufficient. in the context of this work the authors should clearly explain the syntheisis, processing and characteristics of this material in comparison with the numerous other TiO2 photocatalysts that have been reported in the literature. In particular the phase composition (anatase / rutile) and surface area should be reported.
Author Response
- The introduction is well written. However, TiO2 materials as anti-bacterial photocatalysts have been studied for over 30 years. Thousands of publications have been written on the topic. Authors must clarify how this work is novel, and explain the limitations that have prevented TiO2 being used as a antibacterial photocatalyst in commercial applications so far.
Response: We introduced supplementary explanations in the Introduction part regarding the novelty of the study and the limitations that have prevented TiO2 being used as an antibacterial photocatalyst in commercial applications.
- Copper doped TiO2 has been often used in studies of photocatalytic materials. Authors should refer to the previous studies that have used Cu doped TiO2 films for the photocatalytic degradation of methylene blue in thin film form. How does copper affect the anatase to rutile phase transformation?
Response: For the photocatalytic degradation of methylene blue, we referred only to the ISO standard, because being a product that is intended to be passed from the scientific research to the practical application, it must comply with the standard conditions. The transition from the anatase polymorphic form to the rutile form can be done only by thermal calcination and not by the deposition of copper on the surface of the crystals.
- The authors have not reported the characteristics of their materials in a comparative manner. How were the TiO2 materials fabricated? Simply referring to a romanian patent is insufficient. in the context of this work the authors should clearly explain the syntheisis, processing and characteristics of this material in comparison with the numerous other TiO2 photocatalysts that have been reported in the literature. In particular the phase composition (anatase / rutile) and surface area should be reported.
Response:
The characteristics of this material have been presented in detail, in Chapter 2 “Materials and Methods” and Chapter 3 “Results”, where it is shown that we worked only with the anatase form.
At the time of writing, the synthesis and processing of this material is the subject of several industrial patent applications (PCT WO 2019/074386 A1 equivalent RO 132438 B1; PCT / RO2021 / 05007 equivalent RO - A / 00297/2020; application A / 10018 / 2021) under examination for publication in the official bulletin of WIPO. In accordance with the provisions of the Paris Convention on Intellectual Property, under penalty of forfeiture, we are not allowed to publish the method of synthesis and processing until the date of publication in the WIPO Bulletins. The Paris Convention and international IP bodies only allow and approve the publication of studies analyzing methods for characterizing this product.

This manuscript is a resubmission of an earlier submission. The following is a list of the peer review reports and author responses from that submission.
Round 1
Reviewer 1 Report
The purpose of the paper is to analyze the efficiency of photocatalytic protection of a new photocatalytic paint ALINNA (patent RO 132438 B1), based on resins containing photocatalytic pigment (copper-doped TiO2).
The study has a good degree of novelty, yet it sounds confused as it concern the discussion of the results and needs major revisions before to be published in Molecules.
Introduction Section
-Pag.2 #58: please move ‘healthcare associated infection’ at line 51 before acronym HAI.
Photocatalytic activity analysis
-Pag.6: The paragraph is confused in presentation of experimental results, it should be rewritten. In all figures x label is missing and Fig. 1a seems identical to Fig.3, please verify.
Scanning Electron Microscopy
Has the EDAX analysis been performed on the sample area? If yes, a punctual analysis should be performed to verify the nature of plates visible in the centre of SEM micrograph.
Antimicrobial tests
-Pag.14: Figure 14a should be removed, as it is not necessary since results were not statistically significant.
-Pag.15 #434: please specify in the text the cell suspensions used. (Staphylococcus)
Discussion
-The statements reported from #478 of Pag.17 to #509 of Pag.18 should be moved in the introduction section
-As general comments: in the discussion section results should be discussed in the same order they appear in the Result section, furthermore theexplicit reference to the figure is often missing, please add for an easy identification of the reader.
-Pag. 19: the DFT simulation should be moved in the Results section.
Conclusions
-Conclusions sound too synthetic.
In both title and conclusions, the authors refer to ‘long-time protection’, actually no results related to antimicrobial action during time were reported, I suggest to investigate it.
Author Response
Dear reviewer,
Thank you for your kindness and patience in reviewing our paper. We accept all the criticisms and observations that you have made on the paper and we thank you for supporting us in improving its scientific quality
Point 1: Introduction Section
-Pag.2 #58: please move ‘healthcare associated infection’ at line 51 before acronym HAI.
Response 1: We reorganized the Introduction section.
Point 2: Photocatalytic activity analysis
-Pag.6: The paragraph is confused in presentation of experimental results, it should be rewritten. In all figures x label is missing and Fig. 1a seems identical to Fig.3, please verify.
Response 2: The paragraph was rewritten and the figures changed.
Point 3: Scanning Electron Microscopy
Has the EDAX analysis been performed on the sample area? If yes, a punctual analysis should be performed to verify the nature of plates visible in the centre of SEM micrograph.
Response 3: The punctual analysis was performed and a suggestive image was added in the article.
Point 4: Antimicrobial tests
-Pag.14: Figure 14a should be removed, as it is not necessary since results were not statistically significant.
-Pag.15 #434: please specify in the text the cell suspensions used. (Staphylococcus)
Response 4: The fig 14 and 15 were removed and introduced in the Supplementary materials; also, I specified in the text the used strain.
Point 5: Discussion
-The statements reported from #478 of Pag.17 to #509 of Pag.18 should be moved in the introduction section
-As general comments: in the discussion section results should be discussed in the same order they appear in the Result section, furthermore the explicit reference to the figure is often missing, please add for an easy identification of the reader.
-Pag. 19: the DFT simulation should be moved in the Results section.
Response 5: We moved the specified paragraph and we reorganized the Introduction section; we reorganized, also, the Discussion section, in the same order they appear in the Result section; all the figures were referred in the text; We would prefer to keep the DFT discussion in Section Discussion as it is directly related to the discussion about SEM and UV-VIS results.
Point 6: Conclusions
-Conclusions sound too synthetic.
In both title and conclusions, the authors refer to ‘long-time protection’, actually no results related to antimicrobial action during time were reported, I suggest to investigate it.
Response 6: The conclusions were reweighted and the expression ‘long-time protection’ has been deleted because it is not supported by the results presented in this article. In situ investigations for ‘long-time protection’ are ongoing and will be published.

Reviewer 2 Report
The paper entitled "New strategies based on light-activated antimicrobial agents as photocatalytic method for long-time protection of surfaces with increased risk of infections" by Mihaescu et al. has been sent to MDPI Molecules.
The current version of the paper is not suitable of publication. English must be improved all over the draft.
Subindexes must be used to describe correctly the chemical formula through text and equations.
Decimal is indicated by "." and not "," in English. Graphics presented include this mistake in the axes.
Resolution of the figures is really poor and many of them do not offer any significant value or information in the scientific type manuscript.
I do not recommend this paper for publication.
Author Response
Dear reviewer,
Thank you for your kindness and patience in reviewing our paper. We accept all the criticisms and observations that you have made on the paper and we thank you for supporting us in improving its scientific quality.
Point 1: The current version of the paper is not suitable of publication. English must be improved all over the draft.
Response 1: The authors completely reorganized the content of the paper, taking into account all the recommendations received from the reviewers and we consider that the quality of the article has been substantially improved.
Point 2: Subindexes must be used to describe correctly the chemical formula through text and equations.
Response 2: We made all the suggested corrections.
Point 3: Decimal is indicated by "." and not "," in English. Graphics presented include this mistake in the axes.
Response 3: We made corrections for all graphics.
Point 4: Resolution of the figures is really poor and many of them do not offer any significant value or information in the scientific type manuscript.
Response 4: We changed the resolution of the figures and we eliminated the not-relevant results.

Reviewer 3 Report
This article explores the use of light-activated antimicrobial surfaces. The experimental work has been professionally and systematically done. The results seem interesting, and there have been various other validating outputs (i.e.; patents, external testing results, etc.). However, despite all the ingredients being present for a potentially good quality output, this article has numerous organizational, structural and stylistic issues. So many issues in fact that in many cases, in my opinion, it requires wholesale re-writing and re-structuring of the material, in many cases. I recommend major changes, and then later re-appraisal of the manuscript. It is not currently possible to fully consider he quality of the findings due to the unclear presentation. I do not recommend publication in its current form. Certain issues and tips are highlighted below:
ENGLISH:
- Many scientific notations lack the required subscripts, superscripts, etc.
- The article is riddled with typos, too many to enumerate. I strongly recommend a thorough check of the entire manuscript. Indeed, it is recommended that the multiple authors make an effort to proof-read their work prior to submission. Some examples; misspelling of clusters, patent, etc.
- Sentences are often unclear and lack specificity.
ARTICLE BODY:
- The introduction (and indeed, the whole article) is confusing and unclear. It also lacks focus and organization. It is full of general comments, and meanders its way across multiple, tangentially-related topics. Some very basic. Some irrelevant. I recommend completely re-writing the entire article.
- There does not seem to be any meaningful comparison of the experimental result against past literature. It is hard to gauge the relevance and efficacy of the reported system without this context being provided.
- Whilst I have no objection to the judicious use of bullet points, they look very messy and disorganized in this current submission
- Some of the information in the main article can be removed to a supplementary information section. This should help tighten the focus for the main article.
FIGURES & TABLES:
- Most of the figures are at an insufficiently high resolution (e.g.; Fig 2, 10, 11, 13, etc.). Please reformat all figures to a higher output quality.
- All figure labels require greater clarity and detail. E.g.; The table 2 description has little value at present.
REFERENCES:
- The referencing seems a little inconsistent (e.g.; inconsistent capitalization, inconsistent journal abbreviation styles, etc.). In addition, several details seem to be missing form many of the non-journal article references.
- What is the value of referring to the patents multiple times? If they are cited in the references, there is little need to duplicate their listing in another section.
- The light-activated paints and surfaces region is a growing one. The authors may wish to refer to one or more of the following references (or other related ones) to improve their reference list, as well as look to them to better inform the direction and structure of this current article:
- https://doi.org/10.1039/C5RA01673H
- https://doi.org/10.1021/acs.langmuir.8b03584
- https://doi.org/10.1038/s41598-017-15565-5
- https://doi.org/10.1007/s10971-018-4714-y
Author Response
Dear reviewer,
Thank you for your kindness and patience in reviewing our paper. We accept all the criticisms and observations that you have made on the paper and we thank you for supporting us in improving its scientific quality.
Point 1: ENGLISH:
- Many scientific notations lack the required subscripts, superscripts, etc.
- The article is riddled with typos, too many to enumerate. I strongly recommend a thorough check of the entire manuscript. Indeed, it is recommended that the multiple authors make an effort to proof-read their work prior to submission. Some examples; misspelling of clusters, patent, etc.
- Sentences are often unclear and lack specificity.
Response 1: The authors completely reorganized the content of the paper, taking into account all the recommendations received from the reviewers and we consider that the quality of the article has been substantially improved.
Point 2: ARTICLE BODY:
- The introduction (and indeed, the whole article) is confusing and unclear. It also lacks focus and organization. It is full of general comments, and meanders its way across multiple, tangentially-related topics. Some very basic. Some irrelevant. I recommend completely re-writing the entire article.
- There does not seem to be any meaningful comparison of the experimental result against past literature. It is hard to gauge the relevance and efficacy of the reported system without this context being provided.
- Whilst I have no objection to the judicious use of bullet points, they look very messy and disorganized in this current submission
- Some of the information in the main article can be removed to a supplementary information section. This should help tighten the focus for the main article.
Response 2: The paper content was re-writhed and re-structured, starting with introduction section, improving also the references, eliminating the bullet points, removing to the supplementary information section the less-relevant results. We tried to improve the discussion section by comparing similar experimental studies, but also keeping the originality of the basic idea of the photocatalytic paint that was patented.
Point 3: FIGURES & TABLES:
- Most of the figures are at an insufficiently high resolution (e.g.; Fig 2, 10, 11, 13, etc.). Please reformat all figures to a higher output quality.
- All figure labels require greater clarity and detail. E.g.; The table 2 description has little value at present.
Response 3: We improved the resolution of the figures ant the content of the table 2 (in the present form of the paper it has become table 3).
Point 4: REFERENCES:
- The referencing seems a little inconsistent (e.g.; inconsistent capitalization, inconsistent journal abbreviation styles, etc.). In addition, several details seem to be missing form many of the non-journal article references.
- What is the value of referring to the patents multiple times? If they are cited in the references, there is little need to duplicate their listing in another section.
- The light-activated paints and surfaces region is a growing one. The authors may wish to refer to one or more of the following references (or other related ones) to improve their reference list, as well as look to them to better inform the direction and structure of this current article:
- https://doi.org/10.1039/C5RA01673H
- https://doi.org/10.1021/acs.langmuir.8b03584
- https://doi.org/10.1038/s41598-017-15565-5
- https://doi.org/10.1007/s10971-018-4714-y
Response 4: The References section has been improved, taking in consideration the suggested paper (reference 19 and reference 48). We eliminated the Section 6 (Patent) and we kept citing patents for reference only.

Round 2
Reviewer 1 Report
The authors modified the work taking into account all requests. I suggest to accept it in the present form.
Author Response
Dear reviewer,
Thank you for the final appreciation of the paper and all the suggestion that helped us to improve its scientific quality.
Reviewer 2 Report
Despite the implemented changes, I still consider this manuscript not suitable for publication
Author Response
Dear reviewer,
We appreciate that you have noticed that the changes you requested in the 1st round of revision have been made. Since you did not request anything else in round 2, please find on the MDPI platform the revised manuscript, updated according to the observations of the other reviewers.
Reviewer 3 Report
The manuscript is much improved but still has many errors/missing information (methodological, explanatory, English language, etc.). For example:
- UV-vis: what was the step-size for acquisition?
- Photocatalytic testing: what was the distance between the excitation source and the sample(s)?
- There is a lot of supposition in the article - it would serve the authors better to stick to what can directly be proven with data. For example, the Stark-Einstein Law may well apply to the experiment. But the authors do not have any direct data to prove/disprove any claim. So it would seem unnecessary to expound on it.
- Whilst I will leave the details of the computational modelling to a specialist reviewer, I would query the current model. The experimental work spans a concentration range (according to EDX) of 0.35 - 7.84 % copper. This is a very large range over which to assume that defect type would stay constant; the energies usually change in response to conditions (e.g.; dopant concentration). As such, using a universal model to explain bonding interactions for the entire range of experimentally synthesised samples seems inaccurate.
Author Response
Dear reviewer,
Thank you for the second appreciation of the paper and for the suggestions that helped us to improve its scientific quality.
- Point 1: UV-vis: what was the step-size for acquisition?
Response 1: The spectra were collected under ambient conditions using Specord 250 equipment (Analytic Jena), in the 300-1100 nm range (Δλ= 2 nm, scanning speed = 10 nm/s, integration time = 0.2 s).
- Point 2: Photocatalytic testing: what was the distance between the excitation source and the sample(s)?
Response 2: In the paragraph 4.2. are highlighted the parameters used for the photocatalytic test. The distance between the excitation source and the sample is not used because the light power emitted by a source decreases by the square of the distance between the source and the sample. On the other hand, because it is intended that the tested product be certified and produced on an industrial scale, it was necessary to test according to the standard DIN 52980: 2008-10. This standard requires that the irradiance on the sample surface be higher than 10 W / sqm (supplementary material).
- Point 3: There is a lot of supposition in the article - it would serve the authors better to stick to what can directly be proven with data. For example, the Stark-Einstein Law may well apply to the experiment. But the authors do not have any direct data to prove/disprove any claim. So it would seem unnecessary to expound on it.
Response 3: We eliminate all the suppositions that are not directly related with the obtained results.
- Point 4: The experimental work spans a concentration range (according to EDX) of 0.35 - 7.84 % copper. This is a very large range over which to assume that defect type would stay constant; the energies usually change in response to conditions (e.g.; dopant concentration). As such, using a universal model to explain bonding interactions for the entire range of experimentally synthesized samples seems inaccurate.
Response 3: For the morphological analysis of the surface, several areas from the surface of a TiO2 pigment were selected. The concentrations shown in table 1 refer strictly to the areas listed in figure 6 and do not reflect the average concentration relative to the average mass of pigment. Because we are talking about an average for concentration and not particular cases, it is necessary to use a universal model that reflects an average of the presented properties.
The paper does not follow a particular model of TiO2 doped with Cu applied only at the laboratory scale. The TiO2 powder incorporated in the product is for industrial use, in the form of micronized TiO2 granules with dimensions between 220 nm and maximum 4 microns and which (included in the class of "functional pigments"). The article aims to characterize a TiO2 pigment doped with Cu that can be reproduced industrially and widely used. A universal model could present the average of the activities and characterizations of this pigment and to establish certain parameters that must be achieved in the industrial production of this pigment.